# EEG dynamical network analysis method reveals the neural signature of visual-motor coordination

Xinzhe Li[1], Bruno Mota[2], Toshiyuki Kondo[3], Slawomir Nasuto[1], Yoshikatsu Hayashi[1]*

1 Biomedical Science and Biomedical Engineering, School of Biological Sciences, University of Reading, Reading, United Kingdom, 2 Instituto de Física, Universidade Federal do Rio de Janeiro, Rio de Janeiro, Brazil, 3 Department of Computer and Information Sciences, Tokyo University of Agriculture and Technology, Tokyo, Japan

* y.hayashi@reading.ac.uk

**Data Availability Statement:** Data are available from Zenodo (DOI: 10.5281/zenodo.3819640).

**Funding:** The author(s) received no specific funding for this work.

## Abstract

Human visual-motor coordination is an essential function of movement control, which requires interactions of multiple brain regions. Understanding the cortical-motor coordination is important for improving physical therapy for motor disabilities. However, its underlying transient neural dynamics is still largely unknown. In this study, we applied an eigenvector-based dynamical network analysis method to investigate the functional connectivity calculated from electroencephalography (EEG) signals under visual-motor coordination task and to identify its meta-stable states dynamics. We first tested this signal processing on a simulated network to evaluate it in comparison with other dynamical methods, demonstrating that the eigenvector-based dynamical network analysis was able to correctly extract the dynamical features of the evolving networks. Subsequently, the eigenvector-based analysis was applied to EEG data collected under a visual-motor coordination experiment. In the EEG study with participants, the results of both topological analysis and the eigenvector-based dynamical analysis were able to distinguish different experimental conditions of visual tracking task. With the dynamical analysis, we showed that different visual-motor coordination states can be distinguished by investigating the meta-stable states dynamics of the functional connectivity.

## Introduction

Functional connectivity is a widely-used method to understand brain activity under visual-motor coordination [1]. A number of studies have shown that the visual-motor coordination involves the cross-regional correlations of brain cortex [2, 3]. In the work of Roelfsema et.al. [2], cats were trained to perform a visual-motor feedback task in response to visual stimulus, and the cortical local field potentials (LFP) of the animals were recorded. They found beta band synchrony between the visual cortex and the parietal cortex during this visual-motor task. A functional magnetic resonance imaging (fMRI) study by Hamzei et.al. [3] showed that parietal, lateral, and medial frontal brain areas were activated during visual-motor task in

**Competing interests:** No authors have competing interests.

human participants. Mehrkanoon et.al. [4] investigated corticomuscular coherence in the dynamical visual-motor task of human participants. They measured the time-frequency coherency between electroencephalography (EEG) and electromyographic (EMG), and found that the EEG signals were synchronous with the EMG in the beta band, while participants were exerting constant force on the object. This study suggested that neural activities in the alpha and gamma bands involve movement prediction and error correction processes.

In order to define functional networks in brain, the most commonly used measurement methods of neural signals are EEG and fMRI. The fMRI signals have a high spatial resolution, while its temporal resolution is low. In contrast, EEG signals can have the lower spatial resolution than fMRI due to the volume conduction, but its temporal resolution is very high, which makes it an ideal signal acquisition to monitor dynamical properties of the brain. With the EEG signals, different measures of interactions have been used to define the functional connections [5]. Phase-locking value (PLV) is one of the measures that can be used to define the non-directed functional connectivity [6, 7]. Phase-locking is a state of synchrony characterized by two signals keeping constant phase difference for a certain period [8]. It is generally believed that the phase-locking synchrony plays an important role in the neural information transmission process [9–11]. It has been found that degrees of phase synchronization are related to the pathological activity of epilepsy patients [8].

Dynamical functional connectivity is an extension of the functional connectivity with a focus on time sequence of functional connectivity networks. In the context of neural network, the dynamical functional connectivity represents the evolution of neural activity. Many attempts have been made to reveal the dynamical properties of functional connectivity networks [6, 12–15]. One of the ideas is extending the traditional network analysis by considering time development as an additional dimension of the network. In the work of Mattar et.al. [14], virtual links were defined between the nodes on different temporal snapshots of the network. Each snapshot of the network is an instance of the static network, which can be seen as a slice in the network time series. By defining the connections between slices, the dynamical connectivity becomes a 3-dimensional network, with two spatial dimensions and one additional temporal dimension. The community detection methods of static networks can be then extended to this 3-dimensional temporal-spatial network. Another way of introducing dynamical networks relies on partitioning the instances of network into different classes, or states, and then expressing dynamical connectivity as a series of transitions between states. Lehmann et.al. [16–21] have introduced the EEG microstates which utilize the global field power (GFP). The EEG microstates method characterizes the amplitude of EEG signals over the scalp into unique topological patterns of the GFP, which are known as EEG microstates. EEG microstates method focuses on the alpha band signals, and it requires a fitting of the GFP topographies into pre-determined patterns [18].

In this study, we applied the functional connectivity analysis to study the visual-motor coordination [22, 23]. During the visual-motor task, participants were asked to track a moving target with a tracer. This paradigm has also been applied for the investigation of inter-regional interaction patterns of visual-motor behaviour [24–27]. In the work of Rilk et. al. [24], participants were instructed to track an irregularly fluctuating target shown on the screen through a force sensor held by index finger and thumb. They demonstrated that better performances were associated with higher occipitocentral coherence, while high tracking errors were associated with stronger frontocentral coupling.

We used an eigenvector representation of the dynamical functional connectivity to study the evolutional feature of the neural activity. By investigating the eigenvectors associated with the largest eigenvalues of the adjacency matrices, we demonstrated the existence of a metastable states structure of the EEG network in the behavioural tasks. First, we applied the new

dynamical method on a simulated data set, and next, to our EEG data set. At the same time, the standard topological analysis of the functional connectivity network was also applied to compare with the results of dynamical analysis. Our results showed that the eigenvector-based dynamical analysis method was able to extract the evolving patterns of the neural functional connectivity.

## Materials and methods

### Principle of the eigenvector-based method

The eigenvector-based analysis focuses on detecting the phase-locking state of oscillators. In this study, the phase-locking state is defined as a state where the phase difference of two oscillators keeps the same for a certain period:

$$\phi_{m,t} - \phi_{n,t} = \phi_{m,t+1} - \phi_{n,t+1} \tag{1}$$

where $\phi_{m,t}$ is the phase of oscillator $m$ at time $t$, and $\Delta\phi_{m,n,t}$ is the phase difference between oscillator $m$ and $n$ at time $t$. From Eq (1) we can easily obtain

$$
\begin{aligned}
\phi_{m,t+1} - \phi_{m,t} &= \phi_{n,t+1} - \phi_{n,t} \\
\dot{\phi}_{m,t+1} &= \dot{\phi}_{n,t+1}
\end{aligned}
\tag{2}
$$

In practise, we consider two oscillators to be phase-locked if the difference between their angular speeds is smaller than a certain threshold.

It can be also learned from Eq (1) that phase-locking is a continuous state which can not be defined within a single time instance. Thus, the moving time window technique was applied. We calculated the first order differentiation of phases to obtain the instantaneous angular speed and defined a vector for each instance which is called instantaneous angular speed vector. Each entry of the angular speed vector represents the instantaneous angular speed of the corresponding oscillator, as shown on the left side of Eq (3). The moving time window (width = 40 ms, step = 5 ms) was then applied on the angular speed vector time series, and the column angular speed vectors were aligned as following:

$$
\begin{bmatrix} e^{i\dot{\phi}_{1,t_1}} \\ e^{i\dot{\phi}_{2,t_1}} \\ \vdots \\ e^{i\dot{\phi}_{n,t_1}} \end{bmatrix}
\begin{bmatrix} e^{i\dot{\phi}_{1,t_2}} \\ e^{i\dot{\phi}_{2,t_2}} \\ \vdots \\ e^{i\dot{\phi}_{n,t_2}} \end{bmatrix}
\cdots
\begin{bmatrix} e^{i\dot{\phi}_{1,t_w}} \\ e^{i\dot{\phi}_{2,t_w}} \\ \vdots \\ e^{i\dot{\phi}_{n,t_w}} \end{bmatrix}
\rightarrow
\begin{bmatrix} e^{i\dot{\phi}_{1,t_1}} & e^{i\dot{\phi}_{1,t_2}} & \cdots & e^{i\dot{\phi}_{1,t_w}} \\ e^{i\dot{\phi}_{2,t_1}} & e^{i\dot{\phi}_{2,t_2}} & \cdots & e^{i\dot{\phi}_{2,t_w}} \\ \vdots & \vdots & \ddots & \vdots \\ e^{i\dot{\phi}_{n,t_1}} & e^{i\dot{\phi}_{n,t_2}} & \cdots & e^{i\dot{\phi}_{n,t_w}} \end{bmatrix}
\tag{3}
$$

where $t_w$ is the time window width. Within each time window, the row vectors represent the phase trajectories of corresponding EEG channel. Instead of averaging, we calculated the Euclidean distance between each row vector, as shown in the following equation:

$$d_{mn} = \|\vec{v_m} - \vec{v_n}\|_2 \tag{4}$$

where $\vec{v_m}$ and $\vec{v_n}$ stand for the row vectors for channel $m$ and $n$ respectively. The 2-norm of vector difference equals the Euclidean distance between those two vectors. The distance $d_{mn}$ suggests their phase-locking value. Comparing with the moving window averaging, this method preserves all the instantaneous information of the phase time series. By calculating this distance-based PLV for all channel pairs, we obtain an adjacency matrix describing the phase-locking functional connectivity within this time window. After that, a threshold

was applied on the adjacency matrix to determine if certain channel pair is phase-locked, which was determined through the visual inspection. At this stage, an evolving functional connectivity of brain signal was obtained, which was a time series of binarised adjacency matrix.

This PLV measurement is similar to another widely-used technique called phase coherence [28], which is defined as $\text{PLV}_{m,n} = \frac{1}{T} | \sum_{t=1}^{T} e^{i(\phi_{m,t} - \phi_{n,t})} |$, where $T$ is the time window size. However, there are several differences between the phase coherence and the PLV based on the Euclidean distance used in this study. First, the main difference is that phase coherence measures the mean phase difference while our method measures the angular speed difference, which could be more sensitive to synchronization changes than the former in some cases. The other difference is that the projections with the opposite directions of the complex phase difference vector would be offset by each other, and those with the same directions would be enhanced during the calculation of phase coherence, while the Euclidean distance always add the angular speed differences up. For example, an intersection of two phase time series could be incorrectly identified as phase-locked, because the imaginary parts of complex phase difference vector before and after the intersection point would be offset by each other while the real parts would be added up, therefore the final PLV becomes a relatively high value. However, the Euclidean distance is not normalized, which makes it always require a threshold as the benchmark.

The threshold used in this study was determined through visual inspection. We reviewed the raw time series of the angular speed, and calculated the phase locking value for the pairwise signals which were found to be phase-locked in the raw time series. Fig 1 shows a segment of angular speed time series from the actual EEG data, which further demonstrate the selection of the threshold. The two arrow pairs in the figure indicate two example segments of phase-locking. The segments of channel F1 (blue triangles in the figure) and FZ (yellow triangles in the figure) between the magenta arrow pairs are very close to each other during this certain period of time, so they are identified as phase-locked during this period of time. The segments of channel P1 (purple hexagon) and PZ (green hexagon) between red arrows provide another example of the phase-locking channel pairs. With a large number of samples having been reviewed and calculated, the proper threshold ($4 \times 10^{-4}$) for the binary functional connectivity matrix was determined (Varying the threshold did not affect the significance of the differences in the following measures. Please refer to the supplementary material S1 File for more information). At this stage, a time series of binarised adjacency matrix was obtained, which represents the evolving functional connectivity of brain signal.

We call the eigenvector associated with the largest eigenvalue of the adjacency matrix the prime eigenvector. From the time series of binary adjacency matrix, we calculated the prime eigenvector of each time window, generating a time series of prime eigenvectors $\vec{\Phi}(t)$. We conjecture that the evolution of $\vec{\Phi}(t)$ in time represents a trajectory in an N-dimensional dynamical system with a set of at least M attractors given by the stable phase vectors $\vec{\Phi}^1, \vec{\Phi}^2, \ldots, \vec{\Phi}^M$, to be found experimentally. For this picture to be true, even approximately, the system must display a number of testable features:

- The system must spend most of its time close to one of the attractors. Thus, the inner product $\langle \vec{\Phi}(t), \vec{\Phi}(t+1) \rangle \simeq 1$

- If the system undergoes longer excursions away from the attractor, possibly moving into another attractor's basin, self-correlation will decrease $\langle \vec{\Phi}(t), \vec{\Phi}(t+1) \rangle \ll 1$

- In the latter case it will return to the proximity of (the same, or another) attractor.

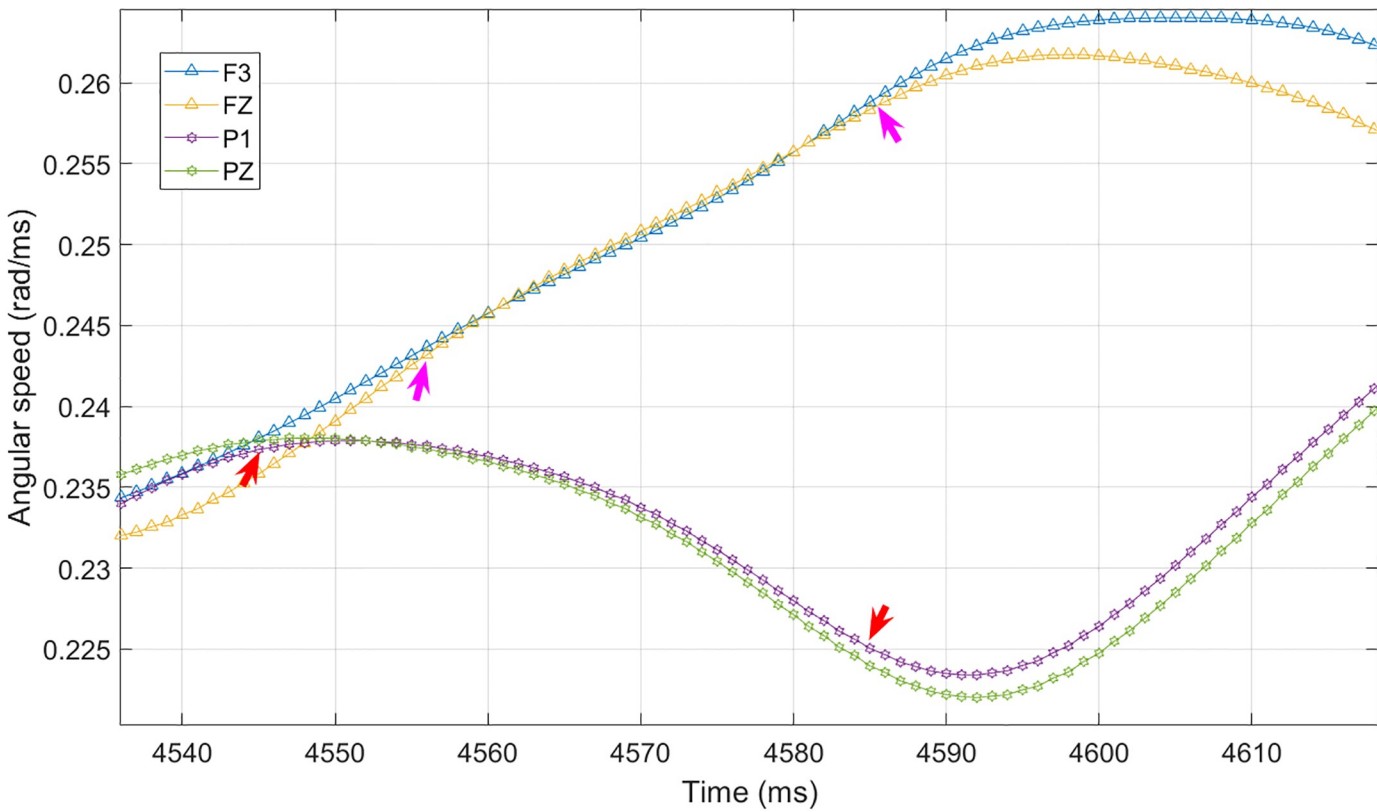

**Fig 1. An example of time series of angular speed, where each curve represents the angular speed calculated from one EEG channel.** The legend shows the location of each electrode in 10-20 system. Phase-locking synchrony suggests the angular speeds of two signals are very close in a certain time period. Therefore in the figure, two signals are considered in a phase-locking synchrony state if those values are within a certain range during that certain time period. Two arrow pairs give two examples of phase-locking synchrony. Channel P1 (purple hexagon) and PZ (green hexagon) are synchronized between two red arrows. Channel F3 (blue triangle) and FZ (yellow triangle) between two magenta arrows give another example of synchronization.

Therefore, investigating the inner product time series of the successive prime eigenvectors can reveal the dynamical feature of the network. We expected that the time series would be a flat line that very close to 1 when the network stay mostly unchanged, and large decreases would take place when major changes of the network happens. In the next section, we set up simulation experiments which provide examples of how this method works, in comparison with other dynamical methods.

## Simulated network

The simulated phase-locking network was built by auto-regression of phase. The phase of an oscillator in the network demonstrates a sinusoidal oscillation with the following rule:

$$\dot{\phi}_{m,t} = \dot{\phi}_{m,t-1} \tag{5}$$

We implemented the phase-locking between oscillators as follows: If one oscillator M is phase-locked with another oscillator N, then the phase increment of M would also affect the phase increment of N. The effect depends on the difference of phase increments, which can be expressed as

$$\dot{\phi}_{m,t} = \dot{\phi}_{m,t-1} + a_{mn}(\dot{\phi}_{m,t-1} - \dot{\phi}_{n,t-1}) \tag{6}$$

where $a_{mn}$ is the strength of the phase-locking. With this method, a phase-locking synchronous network can be simulated. Phases of oscillators were expressed in complex phaser form, then Eq (6) becomes

$$e^{i\phi_{m,t+1}} = e^{i(\dot{\phi}_{m,t-1}+\phi_{m,t})+ia_{mn}(\dot{\phi}_{m,t-1}-\dot{\phi}_{n,t-1})} \tag{7}$$

By defining the initial phase increment and changing the phase-locking strength $a_{mn}$ over time, we are able to simulate a dynamical phase-locking network. In this work, the initial phases are set randomly, and the phase-locking strength $a_{mn}$ is uniformly set as 0.1. The initial phase increment determines the initial position of the oscillator in the angular speed time series, which would be described specifically in each simulation scenario. In the result section, we will describe two different scenarios in order to show the properties of the eigenvector-based method. In each scenario, there were several clusters composed of synchronous oscillators, which would merge or separate from each other during the network evolution. In the following sections, we will show that decreases of the successive prime eigenvector inner product time series occur when clusters merge or separate.

## Comparison with other dynamical clustering methods

Apart from the proposed eigenvector method, two dynamical clustering methods, hierarchy clustering and modularity clustering, were also applied on the simulation data for comparison [29, 30]. The same time series of adjacency matrix which were based on the phase difference were fed into these two clustering methods. For hierarchy clustering, the entries in the adjacency matrix were defined as the distances between the oscillators, and the bifurcation tree was generated from those distances. Then, a threshold of the branches depth was applied on the bifurcation tree, which means that all the bifurcation happened above the threshold would be recognized as different clusters. The value of the threshold was the same as the one applied in the eigenvector analysis.

For modularity clustering [30], we applied two different inputs, one is the weighted adjacency matrix, the other is the unweighted adjacency matrix with threshold. For every time window, a cluster's partition of the network was generated by the clustering method, and then, the cluster's evolution of the network was reconstructed by identifying the overlapping of clusters from different time window.

## Participants of the behavioural experiments

Twelve healthy right-handed participants took part in this experiment. They were all students of the University of Reading, aged between 19 and 24 years old, 4 males and 8 females. The protocol of this experiment was approved by the Ethics Committee of the School of Systems Engineering, University of Reading. All the methods were carried out in accordance with the relevant guidelines and regulations. All the participants were given an introduction to this research and signed a consent form before participating the experiment. The individuals in this manuscript have given written informed consent (as outlined in PLOS consent form) to publish these case details.

## Behavioural task

In the visual-motor tracking paradigm [23], three experimental conditions were included in this study: the Tracking condition (Tra), the Motion Only condition (MO), and the Vision Only condition (VO).

- In the Tra condition, participant sat comfortably in an office chair, using their dominant hand (right hand) to hold a haptic device to control a green tracer on the display (See Figs 2 and 3(a). Informed consent has been obtained from the participant for the publication of identifying images in an online open-access publication). Participants were asked to track a red target, which moved along a circular trajectory with a constant speed (1Hz).

- In the MO condition, participants were asked to move the tracer in a circular trajectory at an arbitrary, but constant speed. The target was not shown on the screen (Fig 3(b)).

- In the VO trials, participants were not taking control of the tracer but passively observing a pre-recorded trial with both target and tracer in the display. The pre-recorded trials were Tra trials that the same participant performed earlier in the experimental session.

The two reference conditions, MO and VO, were designed to decompose the visual and motion components, respectively, from the visual-motion feedback loop. In the MO condition, no target was shown on the display so that participants did not perform tracking activity, while the motion of participants was the same as in the Tra condition and participants received the same sensory feedback as in the Tra condition, which indicates that participants did not have

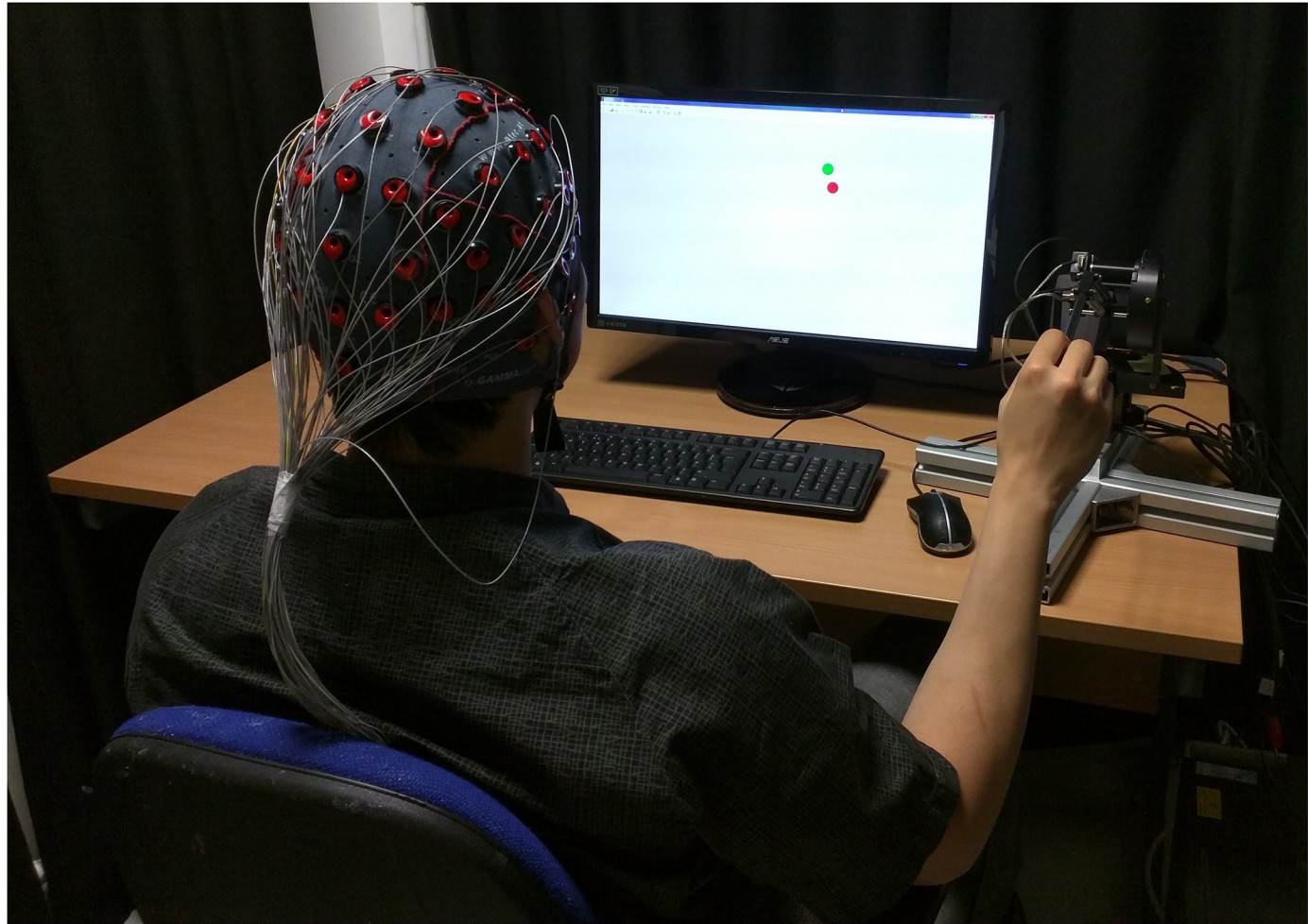

**Fig 2. Photo of the experimental paradigm.** Participant was holding the haptic device to control the tracer in order to track the target. The individual in this manuscript has given written informed consent (as outlined in PLOS consent form) to publish these case details.

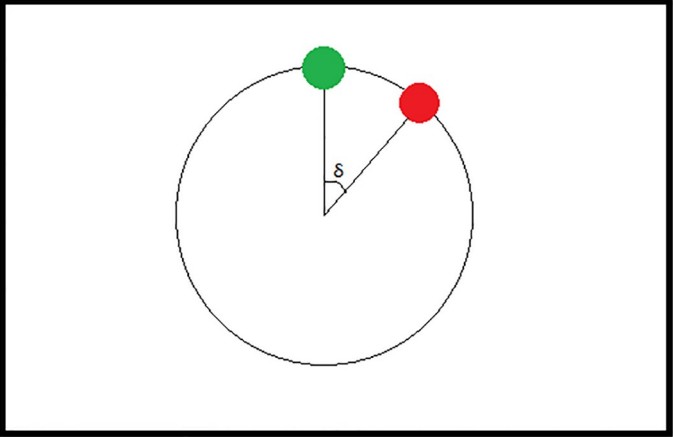
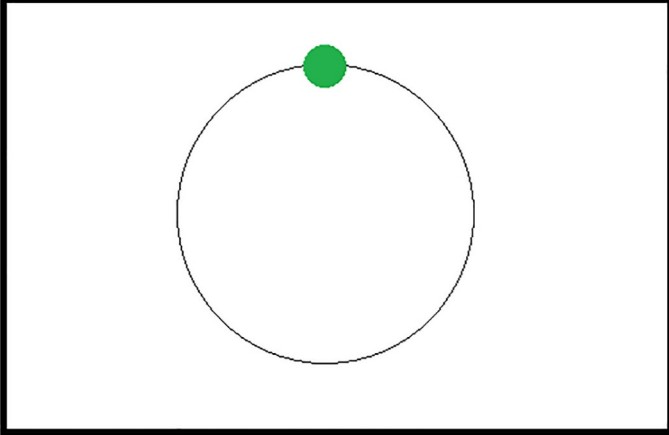

**Fig 3. Display of the experimental conditions.** (a) In Tra trials, participants tracked the red target with the green tracer to minimize the angle $\delta$ between target and tracer. In VO trials, participants passively received the same visual information which were recorded Tra trials. (b) In MO trials, participants were asked to move moved the tracer in a circular trajectory without tracking any target. The black circular trajectory is not visible in the actual experiment.

the visual-based minimization control on their motion. It is necessary to remark the difference between the tracking control in Tra condition and motion control in MO condition.

In the Tra condition, participants were engaged with an intense visual-motor feedback control loop in order to minimize the distance between the target and the tracer, while in the MO condition, the requirement for precision was significantly lower, which resulted in a relatively loose motion control in the MO condition. In the VO trials, participants received exactly the same visual stimulation as in the Tra condition but did not perform any motion control.

By comparing the functional connectivity networks of the Tra condition with those two reference conditions, we were able to separate the neural features related to the vision feedback and the motion control.

Each participant took 20 trials for each condition, which was 60 trials in total. 10 trials made up a set. Each trial lasted for 40 seconds, while a short break of 10 seconds was inserted between individual trials. Participant first performed the Tra trial, then followed by the MO, and finally the VO.

## Signal recording and pre-processing

Electroencephalography (EEG) was recorded through the whole session of the experiment with a 1kHz sampling rate. 32 electrodes (F3, F1, Fz, F2, F4, FC5, FC3, FC1, FC2, FC4, FC6, C5, C3, C1, Cz, C2, C4, C6, CP3, CP1, CPZ, CP2, CP4, P3, P1, PZ, P2, P4, PO3, POz, PO4, Oz) plus 1 reference electrode (FCz) and 1 ground electrode (AFZ) (g.tec) were applied in the experiment. The electrodes were connected to 2 g.BSamp 16-channel amplifiers (g.tec). Each amplifier output the signal to an A/D board (CONTEC CO., Ltd) in order to digitalize the signal, then the digital signal was sent to a PC with xPC-Target (The MathWorks, Inc) running on it. A haptic arm with two encoders, corresponding to the X and Y coordinates of the tracer, respectively, was used for participants to control the tracer. The signals of the two encoders went to a counter board (CONTEC CO., Ltd), which integrated the angle to give the current position of the tracer. Outputs of the counter board were also sent to the target PC. This target PC synchronized the EEG signal and the behaviour signal, then sent to the main PC through an ethernet cable. The programme running on the main PC operated on MATLAB and

Simulink (The MathWorks, Inc). The programme read and recorded data received from target PC, and showed the animation on the display.

MATLAB and EEGLAB (Swartz Center for Computational Neuroscience, La Jolla, CA) were applied to process the EEG data. EEG data were recorded for every experimental trials set. First of all, the datasets were feed into a broad band-pass filtering (0.1-50Hz, all the filtering process in this study applied EEGLAB embedded FIR filter), then the filtered time series were cut into trials. Then an independent component analysis (ICA) was performed on the data in order to remove the eye blinking and other artefacts. We used the EEGLAB embedded function ("runica") to perform ICA. The independent components (ICs) which were recognized as artefacts were selected through visual inspection. There are toolboxes for EEGLAB that automatically perform IC rejection, but they failed to give satisfying result as we tried.

In the manually selection process, the ICs with random occurred stand-alone peaks source from the frontal (blink pattern) and the high frequency jitters (electromyographic pattern) would be considered as artefacts. The whole trial would be rejected if more than 50% ICs were recognized as artefacts. After the ICA pruning, the data were processed with the Laplacian operator through the CSD toolbox [31–33]. This processing was aimed to avoid the phase synchronization resulted from both electrodes sharing a common source.

## Statistical topological properties

We calculated the connection density, mean efficiency and cluster coefficient for each time window [34], and performed time averaging on the series. Instead of the mean path length, we calculated the mean efficiency, which is defined as the averaging of the reciprocal of the shortest path length of channel pairs. The reason for this choice is that the unweighted networks of the EEG functional connectivity were very sparse, with many disconnected nodes, while the shortest path length is not well defined for the isolated nodes. The network efficiency is a generalized metric that describes the same characteristic of network as the shortest path length while it can be applied on the disconnected network. We performed statistical comparisons between the three conditions were performed on the topological properties of all the 12 participants, the result of which is shown in Fig 9.

In order to directly assess the structure of the functional connectivity, the time-averaged functional connectivity (a single network calculated from the dynamical network series) was defined, the connections of which were weighted with their averaged recurrent probabilities. First, the recurrent probability of each connection was estimated within each individual trials. Then the probability networks of all the trials from the same experimental condition were averaged, which made a single representation of the functional connectivity for every condition. In order to demonstrate the network structure in Fig 11, a threshold has been applied to the averaged network. The threshold was chosen to make the most visual differences between conditions.

## Eigenvector based dynamical analysis

In order to extract the mono-frequency oscillators, the narrow band-pass filtering was performed on the data before the Hilbert transform. 4 frequency bands were selected, which are 8-12Hz, 18-22Hz, 26-30Hz, 38-42Hz [35, 36]. After the Hilbert transform, we calculated the PLVs between each oscillators as shown in Eqs (3) and (4) with a moving time window (width = 40 ms, step = 5 ms), and generated an adjacency matrix time series. After that, the matrices were binarized with a visual-based threshold. In order to determine the threshold, we have reviewed the raw phase signal, and calculated the phase locking values for the signal pairs which were found to be phase-locked in the raw signal, as the phase-locking is defined by

constant phase difference. With a large number of samples, we determined the proper threshold ($4 \times 10^{-4}$) for the binary functional connectivity matrix. Finally, the time series of prime eigenvectors was extracted from the time series of adjacency matrix.

We then calculated the prime eigenvectors inner product of two successive in the time series, and compared its distribution between three experimental conditions. The distribution of the inner product time series was found to follow a bimodal distribution, with most of the events fell into either 0 or 1. As discussed in the previous section, the number of 0 in the time series of inner products suggests the number of transitions in the state space. In order to identify the transition frequency, the number of events where the inner product of eigenvectors were $< 0.01$ and $> 0.99$ were counted. Events that $< 0.01$ can be interpreted as prime eigenvector transition between states, and events that $> 0.99$ stand for the situation where prime eigenvector is considered to stay in the same state. The normalized frequency of the two events, $< 0.01$ and $> 0.99$ from each experimental conditions were compared by the Wilcoxon signed-rank test, results of which are shown in Fig 14. Then the distribution of the meta-stable states duration was also investigated.

In order to demonstrate the temporal cluster structure and show the attractors in eigenvector space, we calculated inner product of the prime eigenvectors from all the time windows in the same trial, which generated a correlation matrix. Each entry of this correlation matrix was the inner product of eigenvectors from the corresponding two time windows. To clarify, only a 3-second segment of the whole inner product correlation matrix is shown in Fig 13.

## Results

### Simulation of oscillators for validation of eigenvector based method

The simulated network was set up as described in the Method section. From the simulation results, we found that sharp spikes of inner product correspond to the changes of the largest cluster in the network, such as merging and separation. In order to illustrate how the inner product reacts to the changes of the largest cluster, we present two special scenarios where the changes of the angular speeds of oscillators were set manually, so that the changes of the clusters structure happened instantaneously. After that, we compared the proposed eigenvector-based method with other commonly-used dynamical clustering methods. The phase-locking strength is set uniformly as 0.1. In order to simulate the experiment condition which has a sample rate of 1kHz, each time step is considered as 1 ms.

**First simulation scenario.**  In the first scenario, we present a simple scenario to show the effect of merging and separation of the largest cluster. Two clusters first merged at 500 time step, and then separated at 1500 time step, which is shown in Fig 4. The cluster 1 (marked in blue, 3Hz) had 3 oscillators, while the cluster 2 (marked in red, 5Hz) had 5 oscillators. The simulated network was analyzed with the method described previously, the time window was 40 time step wide, and moved 1 time step each step. The threshold of the adjacency matrix was $4 \times 10^{-4}$. Observing the inner product time series in Fig 4(b), it can be seen that two spikes in the inner product time series appeared at the time step corresponding to the merging and the separation of the clusters, respectively.

To further illustrate how the spikes correspond to the changes to the largest cluster, we showed the prime eigenvector entries in Fig 4(c). In the prime eigenvector before the first spike, the entry indices corresponding to the oscillators in cluster 2 had non-zero absolute values. After the first spike, when the two clusters merged, all the entries had non-zero absolute values. The prime eigenvector after the second spike had exactly the same entries as the prime eigenvector before the first spike. Since the cluster 2 was the largest cluster during 0-500 and

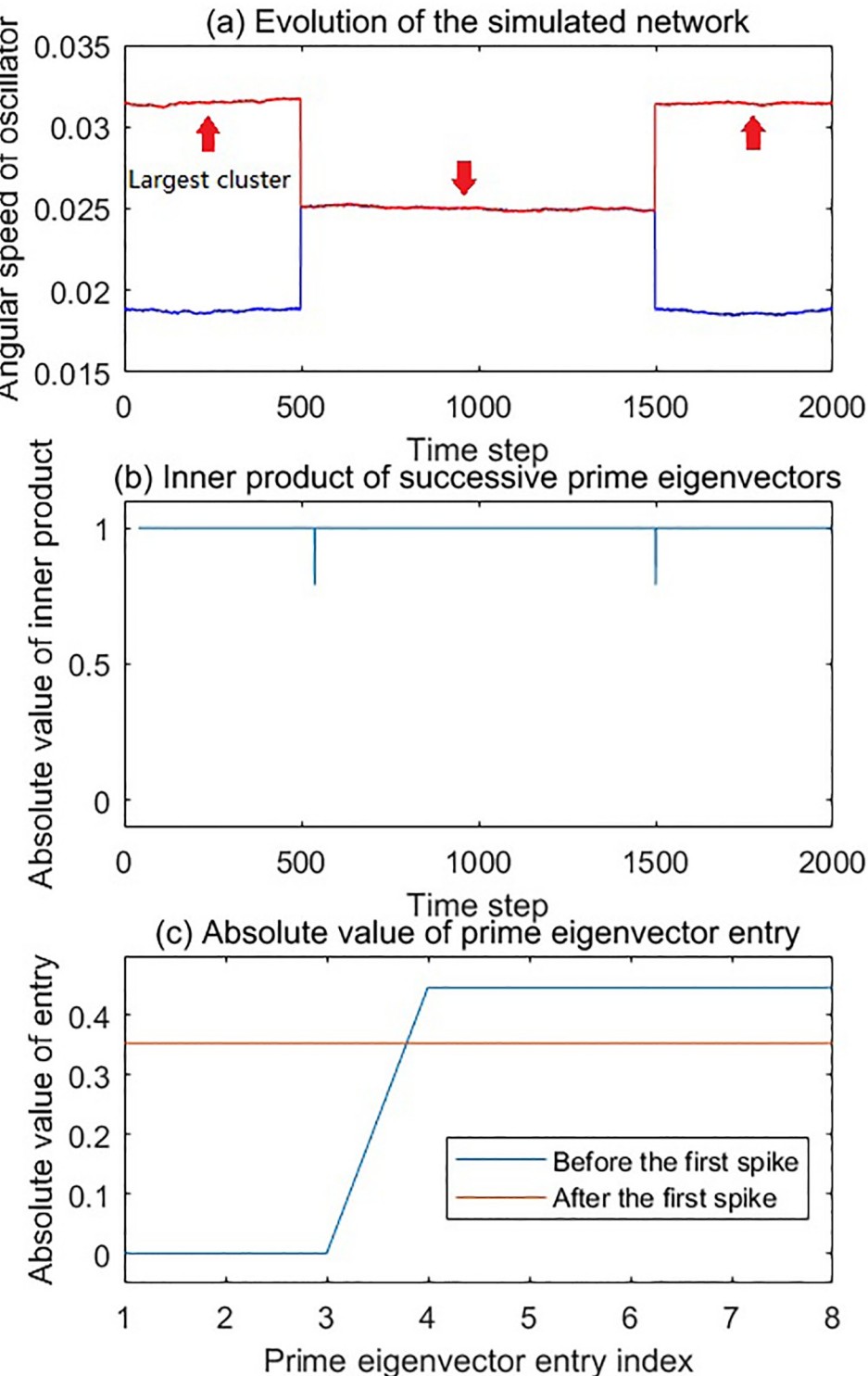

**Fig 4. The evolution and dynamical analysis of the first simulation scenario.** (a) Angular speed time series of oscillators as a function of time. In this simulation, there were two clusters. The oscillators in each of these clusters were shown in the same color. Cluster 1 (blue) had 3 oscillators. Cluster 2 (red) had 5 oscillators. The red arrows in the figure indicate the largest cluster in the network of the moment. At 500 time step, the two clusters were merged. At 1500 time step, the two clusters were separated. (b) The inner product time series of the successive prime eigenvectors has two spikes which correspond to the two network structure changes, respectively. (c) Absolute value of the entry of

two prime eigenvectors. The blue line is the entry value of prime eigenvector selected from the time period of 0-500, while the red line is the entry value of prime eigenvector selected from the time period of 500-1500. During each of these periods, the prime eigenvectors were exactly the same. The prime egenvectors between 1500 and 2000 completely overlap with the blue eigenvector in the figure.

1500-2000, the changes of entry value indicate that the indices of non-zero entries of the prime eigenvector correspond to the indices of oscillators in the largest cluster of the network.

In fact, the work of Allefeld [37] showed that the eigenvectors of the adjacency matrix correspond to the clusters in the network, and their entries indices indicate which oscillator belongs to which cluster. Our finding in this simulation confirmed that the entries of the prime eigenvector provide the information of the oscillator memberships related to the largest cluster of the network. In our analysis, the prime eigenvector was normalized, so the values of the non-zero entries were identical and adopted to the total number of those non-zero entries. Therefore, the inner product of the successive prime eigenvectors drops from 1 if one or more oscillators join or leave the largest cluster.

**Second simulation scenario.** In the second scenario in Fig 5, we show a case where a group of oscillators separated from one cluster and instantaneously merged with another cluster. There were two clusters in the beginning of the simulation. Later at 1000 time step, a part of the largest cluster, which is marked in red in the figure, separated from the the green part, and then joined the blue part. Before the changing event, the cluster 1 (marked in blue, 3Hz) had 3 oscillators, and the cluster with a mixture of red and green included 12 oscillators (5Hz). Among them, cluster 2 (marked in red) had 5 oscillators, and it later joined the blue cluster 1. The residual green part which is named cluster 3 had 7 oscillators.

The simulated network was analyzed with the method described previously, and the parameters were the same as those in the first scenario. The time window was 40 time step wide and moving 1 time step each step. The threshold of the adjacency matrix was $4 \times 10^{-4}$. It can be seen that there were two spikes, a smaller one followed by a large one, around 1000 time step in Fig 5(b). These two spikes indicate two changes of the largest cluster in the network. Before 1000 time step, the largest cluster in the network was the combination of cluster 2 and 3. The first event was that the cluster 2 separated from the cluster 3. The second event was that cluster 2 joined into the cluster 1. In the actual network evolution, these two events would happen in sequence.

In this simulation, where no relaxation existed, these two events happened simultaneously. However, a time window effect took place which recognized these two events separately and showed two spikes in sequence as an inner product time series. This time window effect can be explained by investigating the PLV which is defined as an Euclidean distance in Eq (4). When a time window covers both parts, before and after the event, the PLV of this time window becomes

$$
\begin{aligned}
d_{mn} &= \sqrt{\left\| \vec{v_{m,\alpha}} - \vec{v_{n,\alpha}} \right\|^2 + \left\| \vec{v_{m,\beta}} - \vec{v_{n,\beta}} \right\|^2} \\
&= \sqrt{d_{mn,\alpha}^2 + d_{mn,\beta}^2}
\end{aligned}
\tag{8}
$$

where $\vec{v_{m,\alpha}}$ is the row vector of oscillator $m$ before the change, while $\vec{v_{m,\beta}}$ is the row vector of oscillator $m$ after the change. From the above equation it can be seen that the square of overall PLV is the square sum of two part, one is the oscillator distance before the change $d_{mn,\alpha}$, the other is the oscillator distance after the change $d_{mn,\beta}$. Before the instantaneous change, the cluster 2 is far from the cluster 1, which means $d_{21,\alpha}$ is large. After the change, however, the

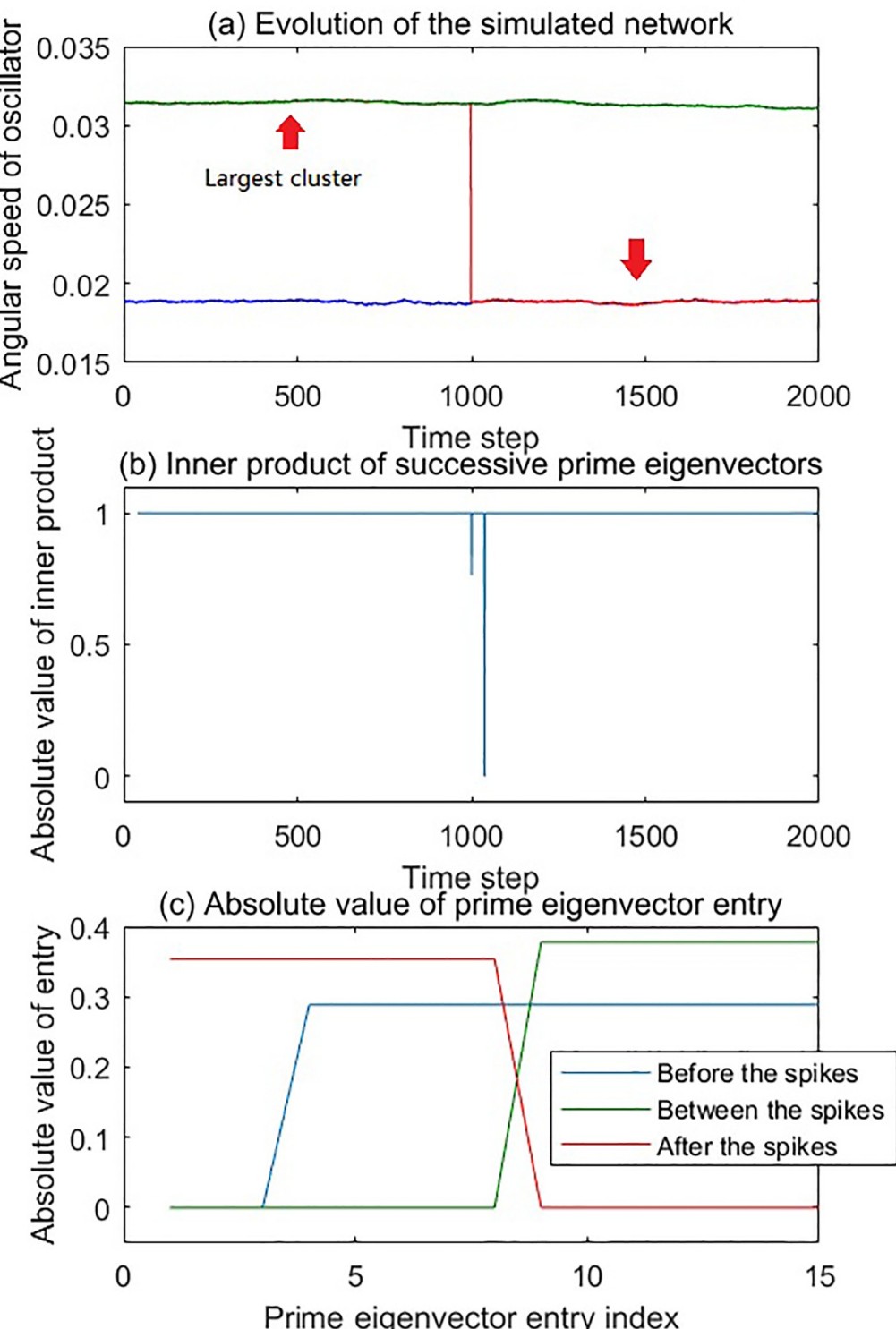

**Fig 5. The evolution and dynamical analysis of the second simulation scenario.** (a) Angular speed time series of oscillators as a function of time. In this simulation, there were three clusters. The oscillators in each of these clusters were shown in the same color. Cluster 1 is in blue has 3 oscillators. Cluster 2 is in red and has 5 oscillators. Cluster 3 is in green and has 7 oscillators. The red arrows in the figure indicate the largest cluster in the network of the moment. In the beginning, cluster 2 and 3 were synchronized while cluster 1 stands alone. At 1000 time step, cluster 2 was separated from cluster 3 and was merged with cluster 1. (b) The inner product time series of the successive prime eigenvectors has double

spikes near 1000 time step. (c) Absolute value of the entry of two prime eigenvectors. The blue line is the entry value of the prime eigenvector selected before the changing event. The green line is selected after but close to the changing event. The red one is selected long after the event.

cluster 2 is fairly far from the cluster 3, which means $d_{23,\beta}$ is large. Therefore, during a certain period when the time window includes both parts around the event, both $d_{21,\alpha}$ and $d_{23,\beta}$ could be large enough to pass the threshold, and the algorithm would identify the cluster 2 as a separate cluster from both cluster 1 and 3. The length of this period is affected by multiple factors, including the length and overlapping of the time window, the threshold of PLV, and the actual relaxation time if the network is not manually manipulated. This feature of the algorithm enables us to separate the two events as if they happened subsequently.

After the separation of cluster 2 and 3, cluster 3 remained the largest cluster in the network, so the actual effect of this event was the largest cluster lost a certain number of members, which leads to a relative small spike. After the merging of cluster 1 and 2, the new combination became the largest cluster in the network, and the cluster 3 was not the largest cluster any more. This change of identity of the largest cluster caused a large spike in the inner product time series. Comparing these two spikes, we found that the depth of the spike in the inner product time series of the prime eigenvector is related to how many different oscillators were in the largest clusters of two different moments, before and after the event.

To further illustrate this point, in Fig 5(c) we show the entries of 3 prime eigenvectors which were selected from different time steps. The blue line is selected from time point that before the network changing event. The green line is selected from time point that after but close to the changing event, where $d_{21,\alpha}$ and $d_{23,\beta}$ are both large. The red line is selected from time point that long after the changing event. It can be seen that the entries with high value correspond to the oscillators in the largest cluster, as we discussed above. In the prime eigenvector before the change, the entries corresponding to cluster 2 and 3 have high values. During the period between the spikes, the high-value prime eigenvector entries correspond to cluster 3 only, because cluster 2 was recognized as a separate cluster. The plateaus of the blue and red line have a certain overlapping, which resulted in a smaller spike in the inner product time series. The merging of cluster 1 and 2 made them the largest cluster, as we can see that the non-zero entries of the prime eigenvector after the double spikes correspond to the oscillators in those two clusters. The non-overlapping shift of the plateau in the prime eigenvector made the inner product 0.

**Benchmark comparison with the dynamical clustering methods.** With the understanding of the nature of the inner product spikes, we then compared our method with the dynamical clustering methods on a simulated dynamical network. Fig 6 shows the angular speed trajectory of the simulated network. The network started with 4 clusters, and there were several merging and separations between these clusters. The oscillators in each of the 4 initial clusters always stay connected during the whole simulation.

The four clusters were named as cluster 1 to cluster 4 from bottom up. Cluster 1 is in blue and has 3 oscillators (3Hz); cluster 2 is in red and has 5 oscillators (4Hz); cluster 3 is in green and has 9 oscillators (5Hz); and cluster 4 is in yellow and has 10 oscillators (7Hz). Oscillator 1-3 belonged to cluster 1; oscillator 4-8 belonged to cluster 2; oscillator 9-17 belonged to cluster 3; oscillator 18-27 belonged to cluster 4.

At time step 500, clusters 1 and 2, clusters 3 and 4 merged together, respectively. At time step 1000, clusters 3 and 4 stopped interacting with each other, while a connection established between clusters 2 and 3. This connection made clusters 3 join the combination of cluster 2

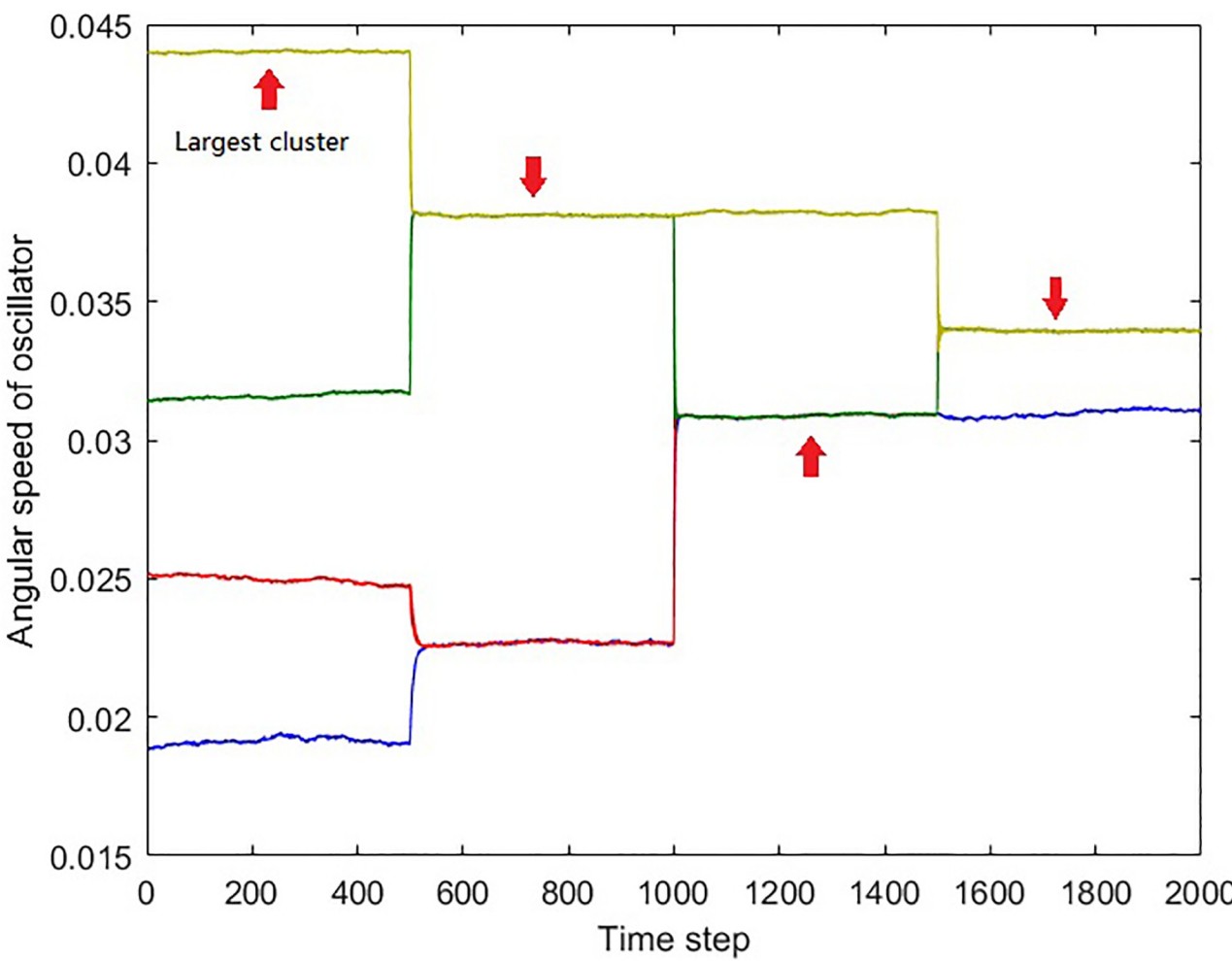

**Fig 6. The angular speed time series of the oscillators demonstrate the network evolution.** In this simulation, there were three non-splittable clusters. The oscillators in each of these non-splittable clusters were shown in the same color. Cluster 1 is in blue and has 3 oscillators. Cluster 2 is in red and has 5 oscillators. Cluster 3 is in green and has 9 oscillators. Cluster 4 is in yellow and has 10 oscillators. At time step 500, clusters 1 and 2, and clusters 3 and 4 were synchronized. At time step 1000, cluster 3 went out of synchrony with cluster 4, but started to synchronize with cluster 2 instead. At time step 1500, clusters 2 and 3 stopped synchronizing with cluster 1 and merged with cluster 4. The red arrows indicate the largest cluster at the moment.

and 3. Finally at time step 1500, clusters 2 and 3 stopped synchronizing with cluster 1 and merged with cluster 4.

Fig 7 shows the time series of the successive prime eigenvector inner product. The parameters of the analysis were the same as previous simulations. It can be seen that the spikes of the inner product correspond to each of the structural changes of the largest cluster in the network. At 500 time step, the largest cluster, cluster 4, merged with cluster 3, caused a small spike in the inner product time series. At 1000 time step, along with the change of network structure, two spikes appeared in the inner product time series. At 1500 time step, cluster 1 separated from cluster 2 and 3, and connected with cluster 4, which caused a single spike in the inner product time series. The generation of the double spikes near 1000 time step is a result that both of the separation and the merging involved the largest cluster. After the separation between cluster 3 and cluster 4, cluster 4 remains to be the largest cluster until cluster 3 merged with cluster 1 and 2. Therefore the separation caused the first smaller spike. The combination of cluster 1, 2, and 3 became the largest cluster after the merging, so the identity

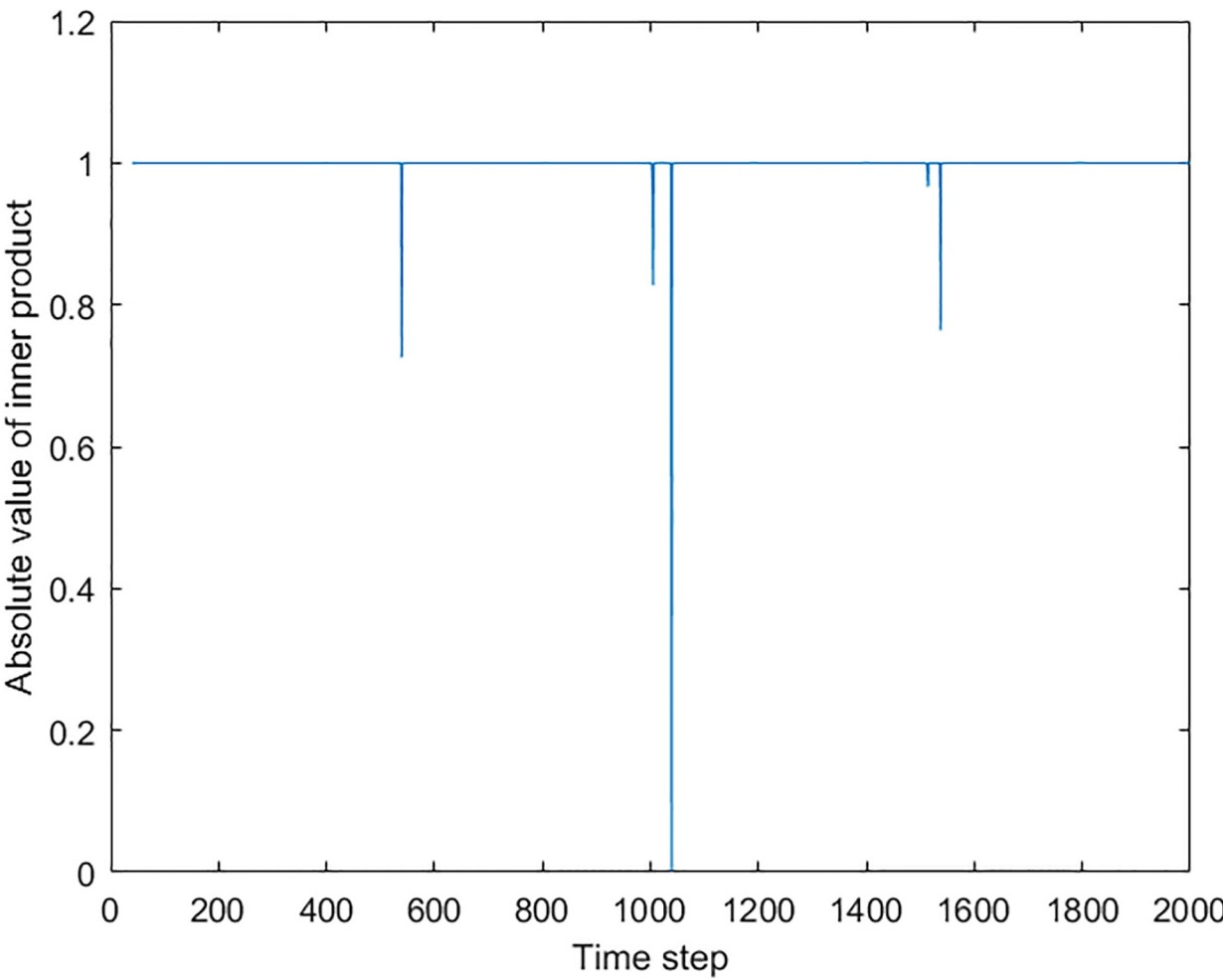

**Fig 7. The inner product time series of successive prime eigenvectors.** The spikes correspond to the cluster changes in the network. At 500 time step, the merging between cluster 3 and 4 caused a spike. At 1000 time step, the change of cluster 3, which separated from cluster 4 and joined cluster 1 and 2, caused double spikes because both of the separation and merging involved the largest cluster. At 1500 time step, clusters 2 and 3 stopped synchronizing with cluster 1 and merged with cluster 4. The separation of cluster 1 made a small spike and the merging with cluster 4 created a larger spike.

change of the largest cluster from cluster 4 to the 1-2-3 combination results in the second larger spike. Also at 1500 time step, The separation of cluster 1 made a small spike and the merging with cluster 4 created a larger spike.

Fig 8 shows the results of the dynamical clustering methods. The vertical axis refers to the indices of oscillators, while the horizontal axis refers to the time step. The colors indicate the indices of different clusters. Subplot (a) shows the result of hierarchical clustering, while subplot (b) and (c) show the results of the modularity clustering. In (b), the modularity clustering was applied on the weighted adjacency matrix. In (c), the modularity clustering was applied on the thresholded adjacency matrix, the same one used in the prime eigenvector analysis. It can be seen that the hierarchical clustering and the modularity clustering with the thresholded matrix successfully captured the cluster changes of the simulated network. However, with the weighted adjacency matrix, the modularity clustering method did not detect the correct cluster structure in the network. Fig 8(b) shows that the modularity clustering method could only distinguish the large gaps before the 1500 time step, such as the gap between cluster 4 and other

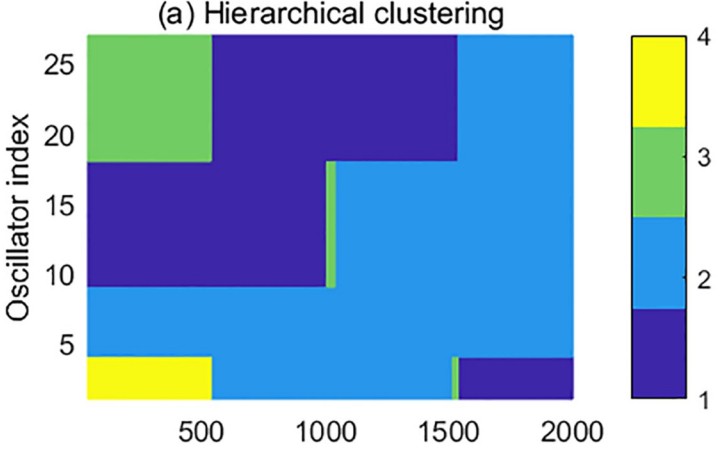

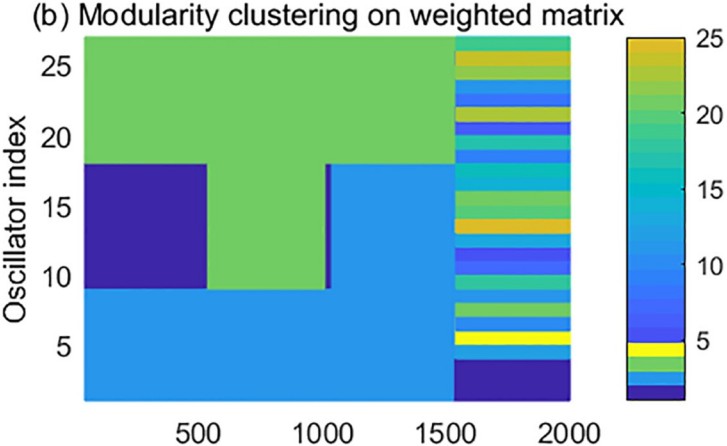

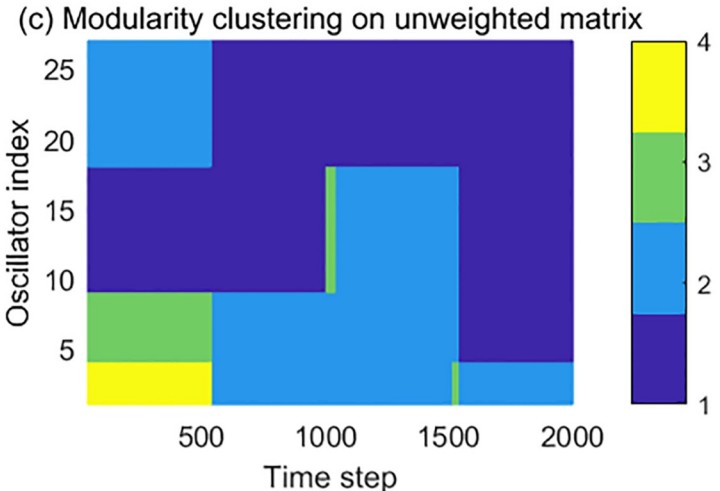

**Fig 8. The evolution of the simulated network structure detected by the dynamical clustering methods.** (a) the result of hierarchical clustering. (b) the result of the modularity clustering applied on the weighted network; (c) the result of the modularity clustering applied on the unweighted matrix. The vertical axis refers to the index of oscillators, while the horizontal axis refers to the time step. Different colors identify different clusters.

clusters between 0 and 500, as well as the gap between two hyper-clusters during 500-1500. After the 1500 time step, the clustering algorithm only recognized the isolated cluster 1. Cluster 2 was also recoginised at the very end, as it drifted away from the large after stopping synchronize with the hyper-cluster at 2000 time step.

However, for the hyper-cluster which consists of cluster 3 and 4, the clustering algorithm could not recognize them as a whole cluster, but singled out every oscillator. Two indications can be made from this result. First, we learned that the modularity clustering algorithm shows some "soft threshold" that could be automatically adjusted with the connection weights distribution. The existence of the large gap makes the soft threshold rather large before the 1500 time step, while the reducing of the gaps after the 1500 time step also reduces the soft threshold that distinguishes different clusters. However, this soft threshold does not perform well in this simulated network, which brings us to the second indication. From the result, it can be concluded that a fixed threshold is preferred in this situation.

From the results of the dynamical clustering methods shown in Fig 8, small strips can be observed when the cluster structure changes. These strips are the result of relaxation during the change, where the cluster separating and merging events could cause discontinuity of the cluster recognition. In a complex network where several structural changing events are taking place, it would be hard to track the network dynamics with those clustering algorithms. By comparing the two prime eigenvectors from different moments, however, the changes of the largest cluster can be recovered in detail without losing continuity, as the prime eigenvector always correspond to the largest cluster in the network. In addition, the simple eigenvector representation of the prime cluster enables easy and direct comparison between the networks at different time instants by performing the inner product. For the hierarchical and modularity clustering, however, the complicated clustering information prevents the direct temporary comparison, which makes them not suitable for the analysis of long time series.

The simplified representation of neural signals is not uncommon. One example is the Event-Related Desynchronization (ERD), which is known as the decreasing of the EEG power spectrum intensity around 10Hz, and has been reported to be as the most common neural signature of attention and motor intention [35, 38, 39]. Even though Fourier spectrum itself over frequency range can provide a wide range of cognitive information, ERD, which is a single scaling parameter, has been used as an indicative parameter for motor and cognitive functions. Our eigenvector-based method is designed to provide a single parameter in the temporary domain which can indicate the most prominent feature of the dynamical changes in cluster structure for a given network. The secondary, tertiary clusters can also be tracked based on analysis of primary eigenvectors. In future works, we will explore how to efficiently extract information from multiple eigenvectors.

## Topological analysis

Before applying the dynamical analysis on the neural functional connectivity, we first performed standard topological analysis to establish the baseline. The time-averaged topological properties of the functional connectivity network have been systematically compared between three different conditions: Tracking condition (Tra), Motion Only condition (MO), and Visual Only condition (VO). Three topological properties, network density, mean efficiency, and global clustering coefficient, were calculated for each trial of motor coordinations. Efficiency is defined as the reciprocal of the shortest path length. It is a generalization of the shortest path length which can be used on the disconnected graphs. Here, we use the efficiency instead of the shortest path length to characterize the network.

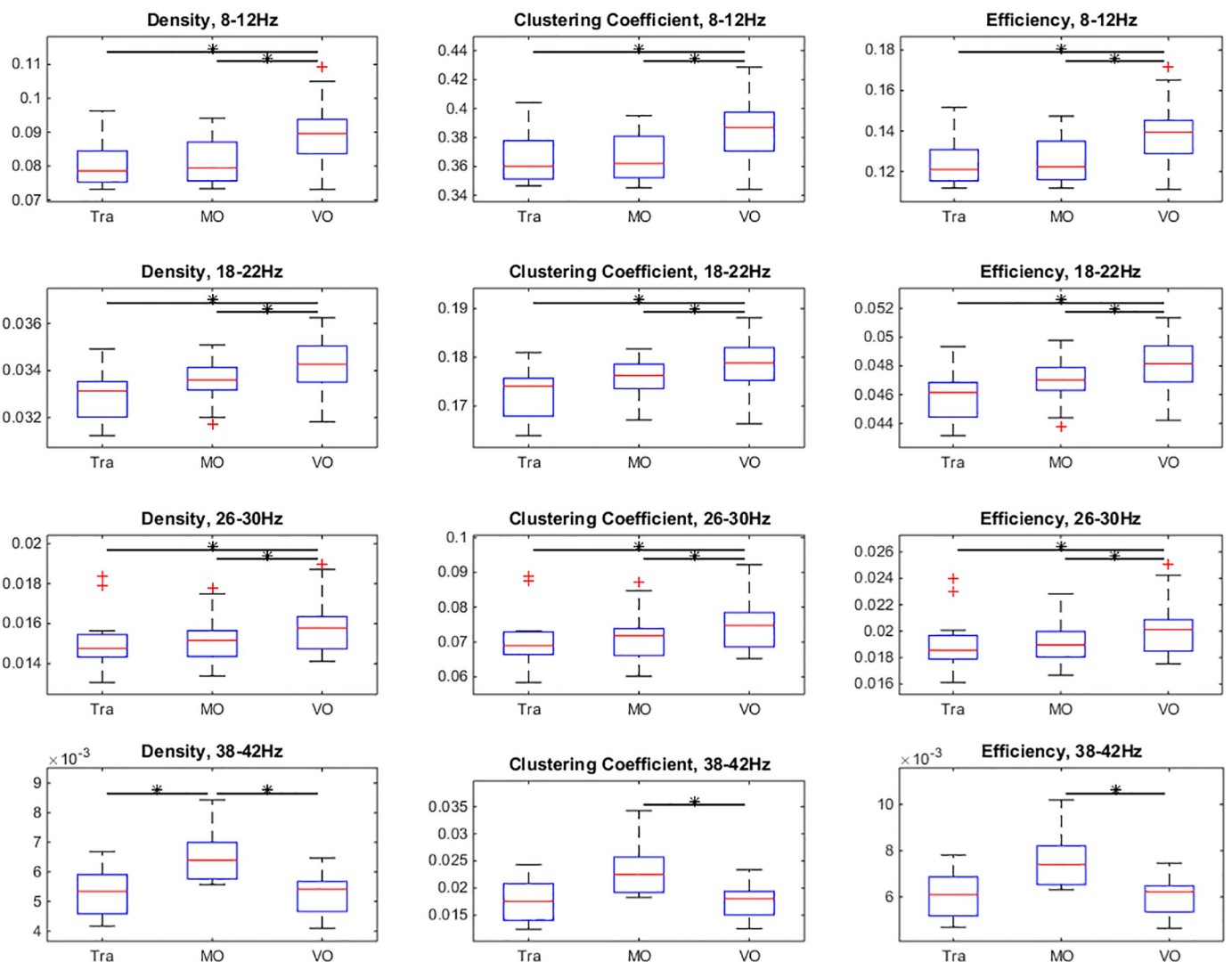

**Fig 9. The box and whisker plot of three statistical topological properties of functional connectivity network.** The values which are more than 1.5 times the interquartile range from the box are marked with red + sign. Each row represents the measurements of one frequency band, while three columns show the three different topological properties of the functional connectivity network. The horizontal bars with stars indicate pairs showing significant difference (paired Wilcoxon signed rank test, $\alpha = 0.05$).

These calculations were performed for four different frequency bands of EEG signals, 8-12Hz, 18-22Hz, 26-30Hz, and 38-42Hz [35, 36]. The statistical test was performed on the time averaged topological properties for all the 12 participants. The results are shown in Fig 9. The paired Wilcoxon signed-rank test ($\alpha = 0.05$) were performed on the data with significant pairs marked with stars.

The common feature found in all the frequency bands was, the three topological properties showed similar patterns in the same frequency band, which means clustering coefficient and the network efficiency are strongly related with the network density. This could result from the sparse network topology. Looking at each frequency band, it can be observed that alpha band (8-12Hz) and the two beta bands (18-22Hz and 26-30Hz) show similar patterns of the topological properties. In these three bands, the VO condition has significantly higher topological properties. From the figure, it can be seen that the differences between VO condition and

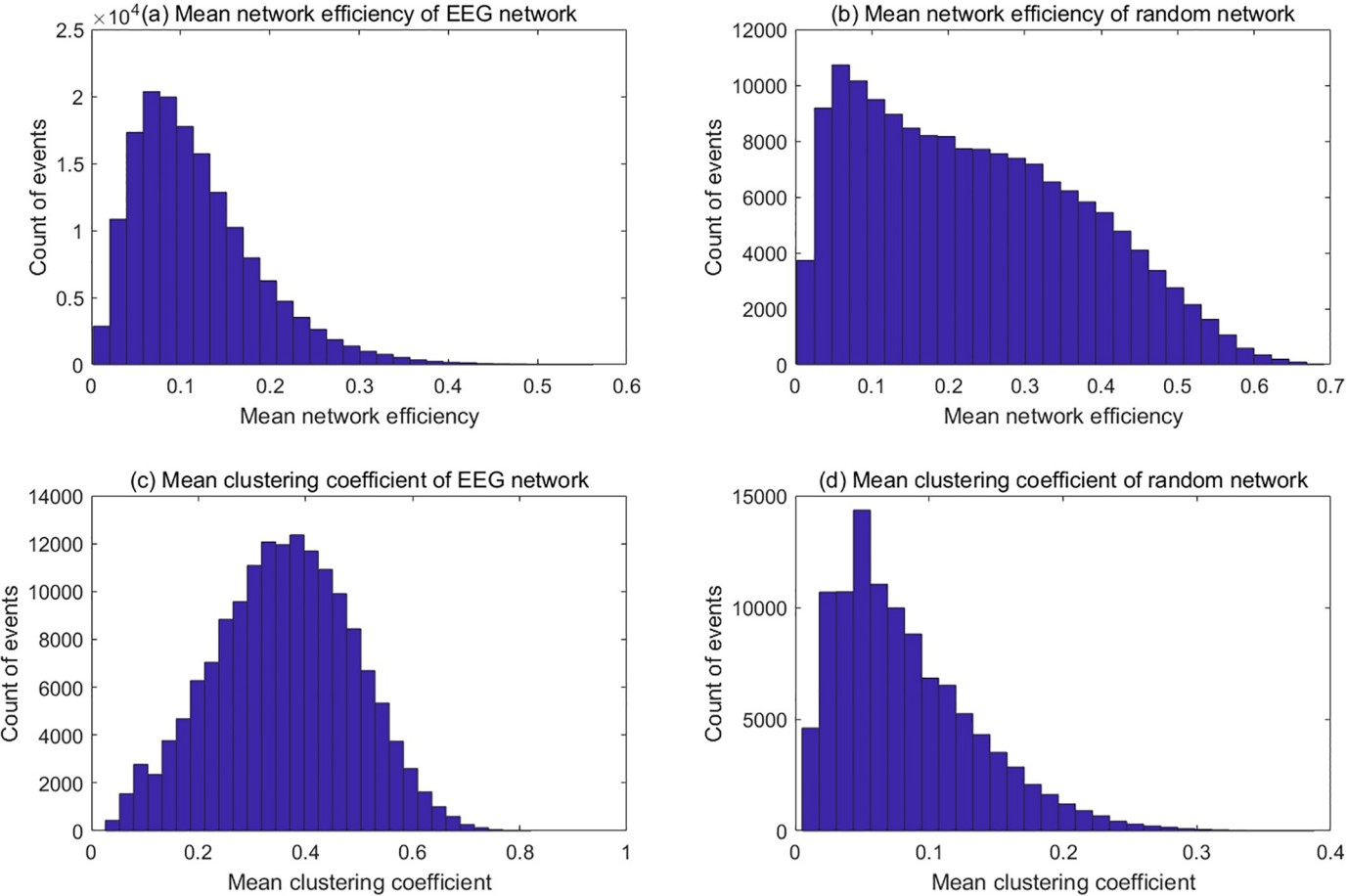

**Fig 10. Distributions of network efficiency and clustering coefficient of EEG functional connectivity and random networks.** (a) Distribution of mean network efficiency of EEG network; (b) Distribution of mean network efficiency of random network; (c) Distribution of mean clustering coefficient of EEG network; (d) Distribution of mean clustering coefficient of random network. The distributions of network efficiency and clustering coefficient were estimated from the tracking trials (Participant ID = dx, 8-12Hz). The random networks were generated with same density of the functional connectivity networks of those trials. The shape of the distributions are different between the EEG networks and random networks. The distribution of network efficiency of random network has fatter tail than that of the EEG network, while the distribution of clustering coefficient is more symmetric than that of the random network. These distributions imply that the connections of the EEG network are more likely to form cliques than the random network, which leads to higher clustering coefficient. That also cause more isolated nodes in the EEG network and lower network efficiency.

the other two condition in the alpha band are the most significant. In the gamma band (38-42Hz), although the differences between Tra trials and MO trials in the clustering coefficient ($p = 0.064$) and network efficiency ($p = 0.052$) is not very significant, the trend that MO condition has higher topological properties than the other two conditions can be clearly observed from the figure.

In order to test whether the clustering coefficient and the network efficiency is only related to the network density, we compared these two properties between the functional connectivity networks calculated from the EEG signals and random networks. We generated random networks whose have the same network density as the EEG functional connectivity. Fig 10 shows topological property distributions of the functional connectivity of tracking trials in alpha band and the random networks with the same density. It can be found from the figure that distributions of both network efficiency and clustering coefficient are different between the functional connectivity and the random network.

The fact that connections are randomly placed in the random network explains that the mean clustering coefficient of random network is smaller than the EEG functional connectivity. It also suggests that there are more isolated nodes in the EEG functional connectivity, which leads to lower efficiency than in the random network, as shown in Fig 10(a). This result shows that although the clustering coefficient and the network efficiency are strongly related with the network density, the EEG functional connectivity network has its own topological structure different from the random network.

In addition to the statistical differences between the global topological properties reported above, we also found differences in the topological structure of functional connectivity network in alpha and gamma bands, which is shown in Fig 11. In order to find the differences of topological structure, we defined an overall representative network for each condition. The connections of these representative networks were weighted by the recurrency probability of each connections in all the trials. For the better demonstration, a threshold was applied on the weighted networks to binarize them.

It is clearly shown in Fig 11 that one of the conditions has a significantly different structure from the other two in each frequency band. In the 8-12Hz band, the VO trials had much denser connections in the occipital part of the scalp. In the 38-42Hz band, the MO trials had more connections than the Tra and VO trials on the right hemisphere and occipital part. In the networks having lower network density, such as the Tra and MO conditions in the alpha

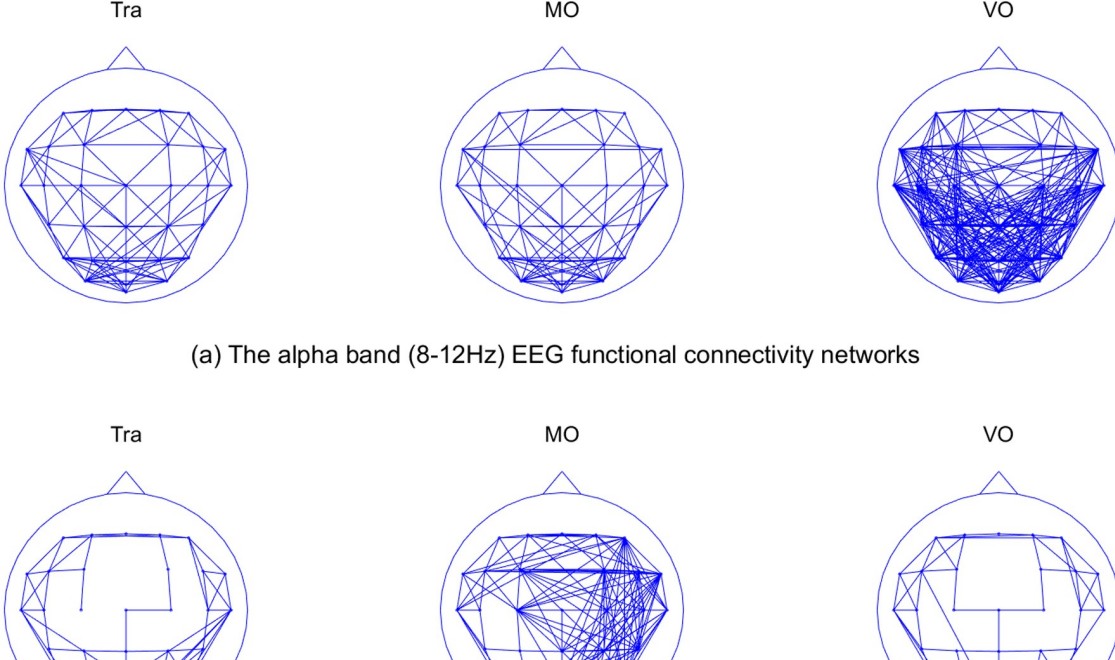

(a) The alpha band (8-12Hz) EEG functional connectivity networks

(b) The gamma band (38-42Hz) EEG functional connectivity networks

**Fig 11. A comparison of functional connectivity structure in alpha and gamma band between conditions.** (a) The functional connectivity network of alpha band (b) The functional connectivity network of gamma band. The probability of each link was calculated for each trial, then the trial-wise averaging was made, which generated a single representative network for every condition. For the purpose of better demonstration, a threshold was applied to the averaged network and it was tuned in the way so that difference of network structure can be clearly shown.

band, we found that their connectivity structures were close to that of the lattice network, which could result from the neighbouring effect of electrodes. Two electrodes may receive the signal from a common source, which then contributes the same component to both channels, which would ultimately result in this channel pair's higher phase-locking value.

## Eigenvector analysis

In order to get an insight into dynamics of the functional connectivity, the evolution of the prime eigenvector of the binary adjacency matrix of the phase-locking network was studied. Following the conjectures made in the Introduction section, we expect that a plot of inner product of successive prime eigenvectors $\langle \vec{\Phi}(t), \vec{\Phi}(t+1) \rangle$ will be composed of long plateaus of high values close to one, punctuated by short transient dips. The set of the average values of $\vec{\Phi}(t)$ over each stable period will be the attractors.

Therefore, we calculated the inner product of successive prime eigenvectors. Fig 12(a) shows the inner product time series of an example trial. This figure indicates that the inner products tend to be either 0 or 1, and their distribution is shown in Fig 12(b). The results shown in these figures are compatible with the noisy attractor dynamics we assumed in the Introduction section. To investigate further, we next investigated the correlations between all eigenvectors in the same time series to see if the attractor structure can be explicitly shown. Fig 13 shows the inner product matrix of a time series segment of the trial in Fig 12, which shows a block structure. The blocks of high value suggest the prime eigenvectors with the similar directions, which can be seen as temporal clusters of the eigenvector trajectory.

Remarkably, the high value blocks on the diagonal stand for the meta-stable states, which correspond to the plateaus of high values in the inner product time series. Fig 13 proves that it would be possible to identify attractors in the prime eigenvector space experimentally. Learning the overall picture of the eigenvector evolution, let us revisit the time series of successive eigenvector inner product. As we assumed previously, $\langle \vec{\Phi}(m), \vec{\Phi}(n) \rangle \ll 1$ suggests a transit a transition between meta-stable states while $\langle \vec{\Phi}(m), \vec{\Phi}(n) \rangle \simeq 1$ can be viewed as holding the same state during that time window. Therefore, the number of event '0' in the inner product time series suggests the number of transitions happened in the state space.

In order to identify the transition from one state to another, we counted the events where the product of eigenvectors were $< 0.01$ and $> 0.99$ (which would be noted as 'Event 0' and 'Event 1' in the following texts) and estimated their frequencies, while events that $< 0.01$ can be interpreted as the prime eigenvector transits between different states, and events that $> 0.99$ stand for the situation where prime eigenvector stays in the same state. We performed significance tests (paired Wilcoxon signed rank test, $\alpha = 0.05$) on the normalized frequency of the two events between different conditions, and the results are shown in Fig 14.

Comparing with the topological properties shown in Fig 9, it can be seen that Tra-VO pair in the alpha (8-12Hz) and Tra-MO pair in the gamma (38-42) are significantly different in terms of both topological and dynamical properties. In the alpha band, frequency of Event 0 of VO trials is significantly smaller than that of Tra and MO trials, while the frequency of Event 1 of VO trials is significantly larger than the that of Tra and MO trials.

These results indicate that the prime eigenvector in VO condition has significantly less transitions than the prime eigenvector in Tra and MO condition in the alpha band. In the gamma band, it can be observed that the MO condition has significant larger Event 0 frequency and smaller Event 1 frequency than those of the Tra and VO conditions, which means that the prime eigenvector of MO condition has significantly more transitions than the other two conditions.

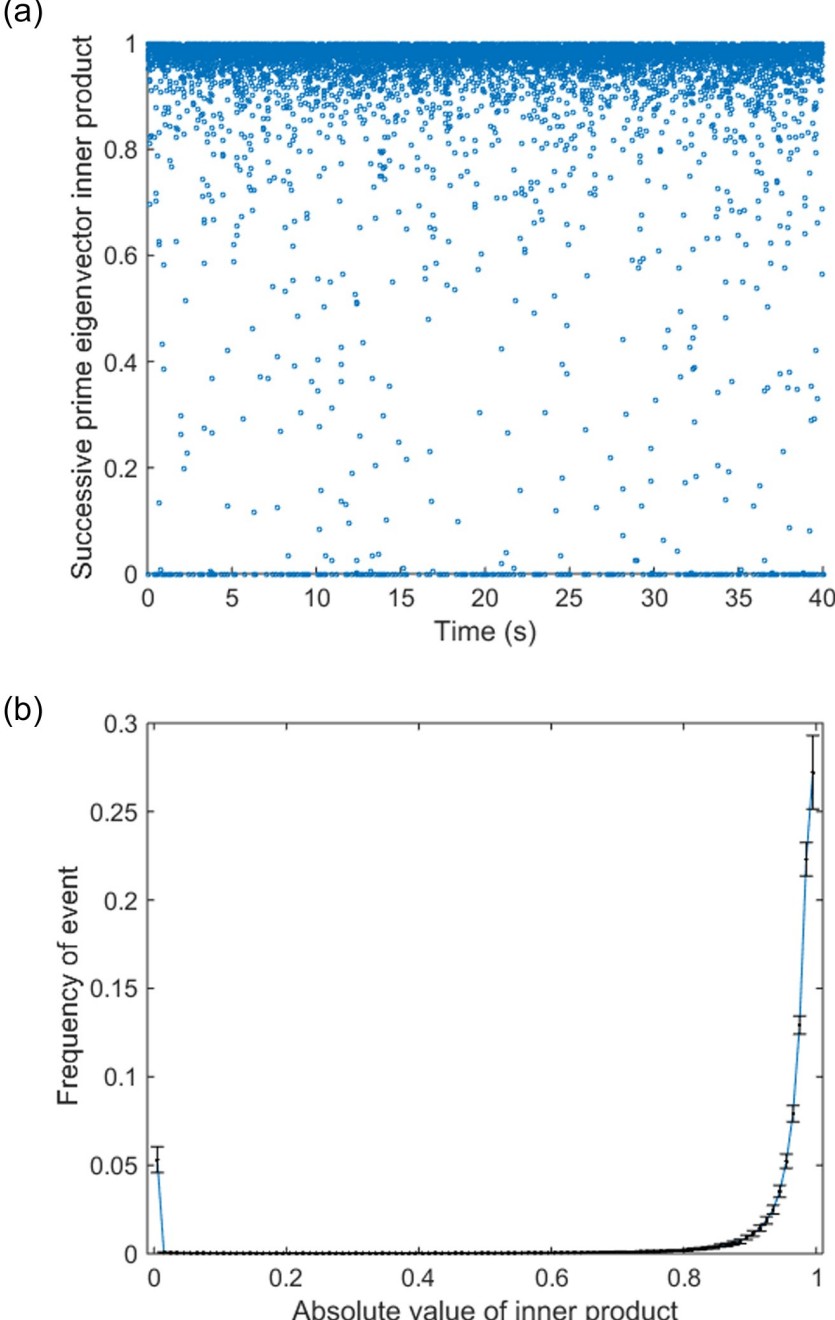

**Fig 12. Time series of successive prime eigenvector inner product.** (a) Example time series of successive prime eigenvector inner product. The figure shows the data from a single tracking trial in alpha band. (b) Distribution of successive prime eigenvector inner product (Tra, alpha band). It is obvious that the distribution follows a bimodal distribution with most of the event counts fall in 0 and 1.

## Discussion

In this paper, we applied the eigenvector-based dynamical method to study the EEG functional connectivity network of visual-motor coordination. We first examined the dynamical network analysis method on the simulated network, whose result showed that this method successfully

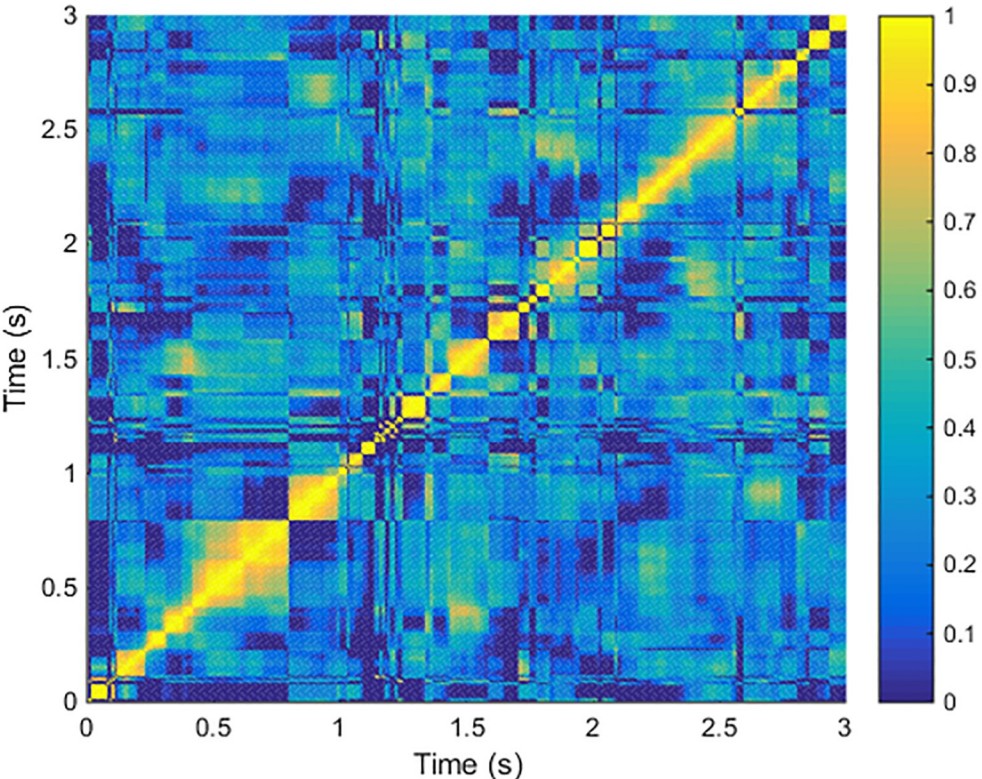

**Fig 13. Inner product matrix of a short segment from the experimental trial shown in Fig 12.** A segment (from 0 s to 3 s) of eigenvector time series was selected, and then inner product was calculated for every possible eigenvector pair in the segment so that each entry in the matrix is the inner product of the corresponding two eigenvectors. It can be observed from the figure that there is a block structure. Two plateaus of the successive eigenvectors inner product time series make a block of high value in the matrix. We can learn that each block refers to a temporal cluster, or an attractor in the eigenvector space. This structure suggests a natural state partition of the meta-stable states.

detected the change of the largest cluster in the network. By investigating the inner-product time series of the neural functional connectivity network, we discovered the meta-stable states structure of the network evolution in the EEG functional connectivity. We estimated the mean frequency of transitions between meta-stable states. By comparing the conditions with respect to the mean meta-stable state transition frequencies, we found that the VO condition in the alpha band had significantly smaller transition frequency between different meta-stable states than the other conditions. In the gamma band, it was also found that the MO condition had significant higher transition frequency than the other two conditions.

Interestingly, significant differences of the topological properties were also found between these conditions in the corresponding bands. We found that mean network density, mean efficiency, and mean cluster coefficient, were able to distinguish VO condition from the other conditions in the alpha band, as well as MO condition from the other conditions in the gamma band. The significantly differences of the functional connectivity networks found across different frequency bands of EEG, especially in alpha and gamma band, indicate that the visual-motor coordination involves information transmission in multiple frequencies. The results also suggest that the eigenvector-based dynamical method is able to distinguish different experimental conditions by revealing the dynamical differences of the network evolution.

Fig 9 provides an overview of the topological properties differentiating between conditions within different frequency bands. In the alpha band (8-12Hz), the VO condition was

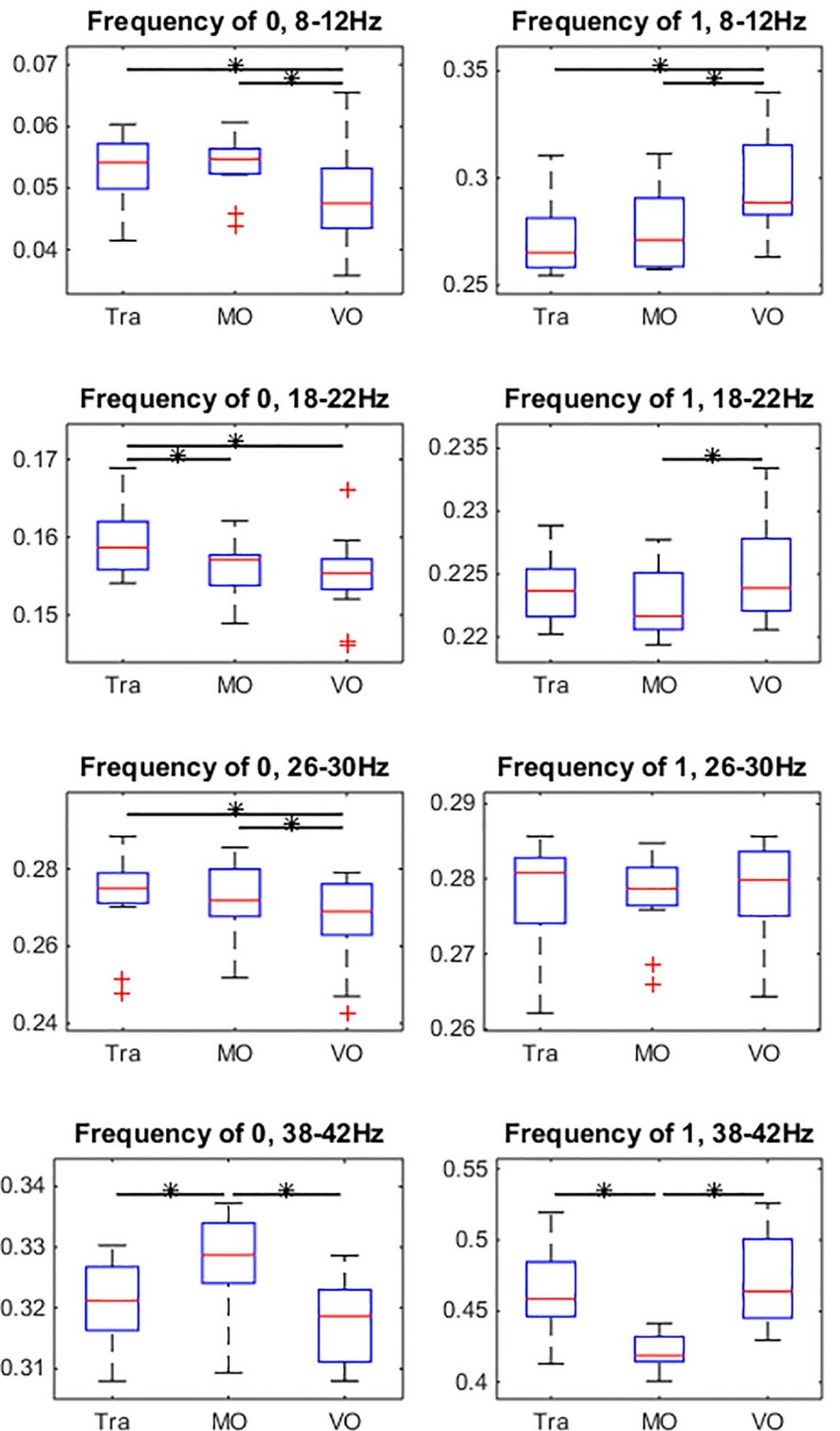

**Fig 14. Comparison of probability of events that $d < 0.01$ (Event 0) and $d > 0.99$ (Event 1), where d is the successive prime eigenvector inner product.** The values which are more than 1.5 times the interquartile range from the box are marked with red + sign. The horizontal bar with star indicates the pair that is significantly different (paired Wilcoxon signed-rank test, $\alpha = 0.05$). This result is highly consistent with the result of topological properties (Comparing to Fig 9).

significantly different from the other two conditions in all the topological properties, which indicates that the alpha band neural activity correlates to the motor control. In the gamma band (38-42Hz), the MO condition was significantly different from the other two conditions, in term of all topological properties as well, which indicates that the gamma band activity correlated to the visual processing and tracking behaviour. The spatial differences of the connections in the functional connectivity network between conditions were also found in both alpha and gamma band, which is shown in Fig 11. The beta band (18-22Hz and 26-30Hz) showed similar patterns of the topological properties as the alpha band, but no significant difference of the topological network structure was found (not shown).

By comparing the topological properties between the EEG functional connectivity and random networks, we showed that the EEG functional connectivity has a special topological structure which is different from random networks, which also implies that the topological properties are not only related to the network density. Fig 9 shows that Tra condition had a lower density of connections in both frequency bands, while the VO condition in the alpha band and the MO condition in the gamma band had a higher connection density. If we consider the VO condition in the alpha band and the MO condition in the gamma band as two baseline conditions for the open visual-motor loop, we can conclude that the closing of the loop during tracking task, Tra, led to desynchronization in both frequency bands. This desynchronization could be related to ERD, which has been reported to be related to the motion control [38, 39]. During the continuous visual stimulus, a series of signals that were time-locked with different instants of visual stimulus were generated, but due to the nature of the continuous events, these signals were not synchronous with each other, which resulted in the desynchronization of EEG.

In this study, we found the inter-channel desynchronization, which may have resulted from the same mechanism: the real-time visual-motor coordination requires intensive communications to establish the fine control, and the neural signals of the communications could be asynchronous due to the random nature of disturbance. Therefore, the time averaged synchrony measurement showed lower value of synchrony.

In the gamma band (38-42Hz) in Fig 11, the MO showed different network structures and topological properties from the other two conditions. In order to interpret this difference, let us revisit the behaviour conditions. In the MO condition, participants were asked to perform a circular motion of the tracer while the target was not shown on the display. In the VO condition, participants were asked to passively watch a pre-recorded tracking trial with both target and tracer shown on the display. As the VO trials exhibited the exactly same visual input as Tra trials, we can conclude that the differences of functional connectivity structures between MO and the other conditions in the gamma band resulted from the different visual inputs from the display. There are two possible explanations of the gamma band neural activity.

One explanation is that participants applied different levels of attention on the moving objects which generated different gamma band connectivity networks. A number of studies have demonstrated that the visual attention is related to the gamma oscillation [40–42]. In the Tra and VO trials, multiple moving objects were shown on the display, which could be more likely to catch participant's attention than showing a single moving object which was under the control of participants.

Another possible explanation is that the different intentions of participants resulted in the differences of functional connectivity in the gamma band. As the participants watching both target and tracer, either with control (in Tra trials) or without control (in VO trials), participants would start to visually measure the distance of the two objects and activated the relative neural circuits. The gamma band functional connectivity features will be further studied in our future work.

From our analysis, we discovered the attractor structure in time domain of functional networks in the prime eigenvector space. As the scale of the eigenvector can always be normalized, its direction is the only property that carries information. So the inner product between two prime eigenvectors can measure the topological difference of the network from different instances. The inner product would have a high value if the network only had minor changes between the two instances, while major changes of the network structure would result in low inner product values. Fig 13 gives a glance at the eigenvector trajectory, which shows many blocks of high values. These blocks are temporal clusters composed of prime eigenvectors with similar directions, which indicates that the network has similar topological structure at these instances. This block structure also suggests that the prime eigenvector trajectory follows a meta-stable dynamics, while the temporal clusters can be interpreted as attractors in the prime eigenvector space. The attractor structure provides a natural partition of the brain states. The continuous 1 series separated by 0 can be viewed as a series of meta-stable states separated by quick but infrequent transitions. The concept of defining state space on the brain signal analysis has been proposed a long time ago. Lehmann and his colleges have proposed the concept of EEG microstates, which is based on different EEG power spatial distribution patterns [16, 17]. There are also other methods based on the clustering of network topological properties [43, 44]. However, the prime eigenvector is a representation of the instantaneous network which preserves the majority of network structure, so it may provides a better way of defining brain signal state space.

By calculating the inner product between the successive prime eigenvectors (i.e., between neighbouring time windows), we can learn how much the network changes between successive instances, and characterize how smoothly the network evolves. As the distribution of inner product follows a bimodal distribution, our focus was on the two major events, corresponding to inner products equal to 1 and 0. Distribution with more Event 1 and fewer Event 0 means that the prime eigenvector has smaller probability to jump between attractors, which suggests that the network is more likely to preserve its structure during the evolution.

From the results (Fig 14), we learned that the VO condition had fewer transition events than the Tra and MO condition in the alpha band, while the MO had larger transition probability between the meta-stable states than that of the Tra and VO condition in the gamma band. It also shows that the network in the lower frequency bands always has lower transition frequency than the network in the high frequency bands. This observation could result from the fact that the phase-locking synchrony changes slower in the low frequency bands, which makes the functional connectivity based on the phase synchrony evolves slower as well.

In addition, it was interesting to find that the condition-pairs showing differences in eigenvector inner-product also showed differences in the topological properties. Comparing Fig 14 with Fig 9, it can be found that trials with significantly different transition frequencies also had significantly different topological properties. This indicates correlations between the network dynamics and static properties of functional connectivity.

The idea of studying the network evolution by the prime eigenvector analysis has been applied before [45]. Cabral et. al. discovered the states transitions in the rest functional connectivity of fMRI signal by similar prime eigenvector representation of the network [46], and showed that the state transitions are related to the cognitive performance of older adults. However, to our knowledge, this is the first study that applied the prime eigenvector method on the EEG dataset, and demonstrates the relation between the meta-stable states of EEG functional connectivity and the visual-motion coordination of human. There are also a number of differences between their techniques and ours. In the paper of Cabral et. al. [46], they used cosine of phase difference as the phase-locking measurement, which can only detect in-phase and anti-phase synchrony.

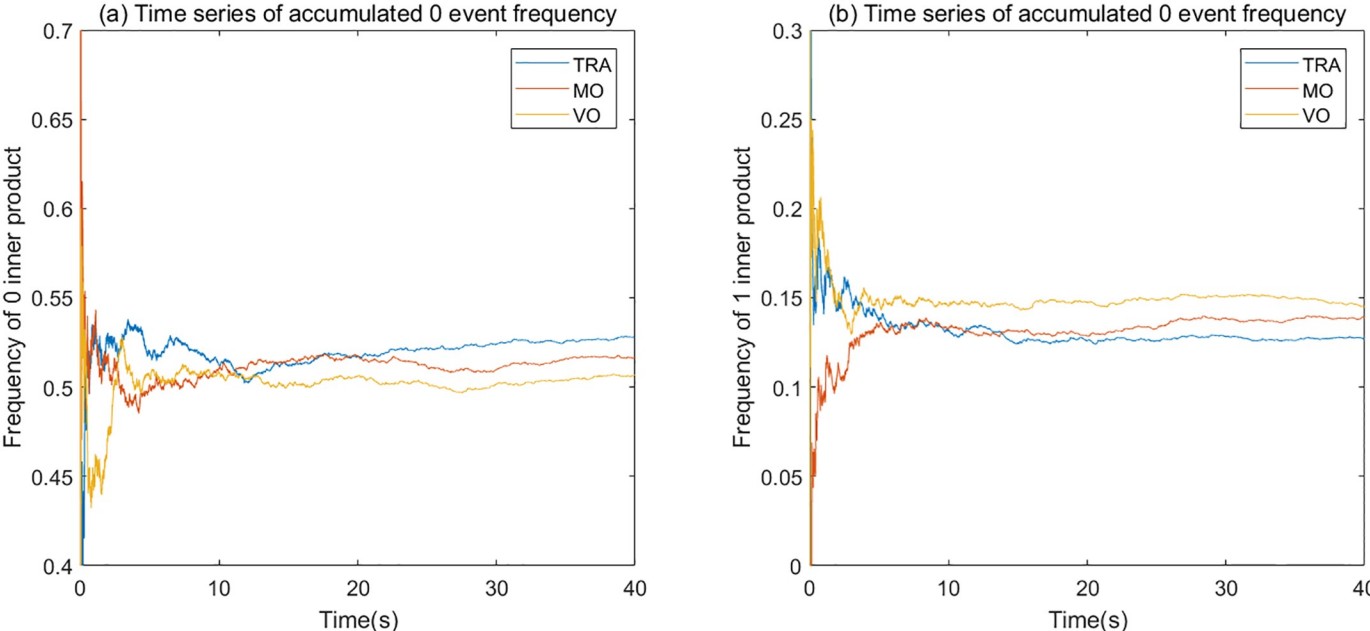

**Fig 15. Accumulated event frequency time series of 3 random selected trials with different experimental conditions.** (a) Accumulated Event 0 frequency; (b) Accumulated Event 1 frequency. All the 3 trials are selected from the alpha band and the same participant (ID = dx). The accumulated Event 0 frequency in the VO condition is smaller than the other two conditions, and its Event 1 accumulated frequency is larger than the other two conditions, which is identical with the statistical results shown in Fig 14.

In our study, we applied Euclidean distance of angular speed vector to measure the phase synchrony by which any constant phase lag would be recognised as phase-locking. As we discussed in the Method section, the angular speed difference could be more sensitive than the phase coherence in some cases, and the property of the Euclidean distance is able to prevent certain type of miss-recognition of synchrony event.

Furthermore, the inner product time series analysis used in this study could be potentially applied on the online processing of Brain Computer Interface (BCI) [47, 48] as a monitoring method to detect meta-stable states. BCI is a system, for example, which can capture brain signals to assist subjects who have movement or communication disability. Therefore, it requires an online brain signal analysis method. While Cabral's work [46] studied each and every eigenvector of network snapshot through the whole time scale, by focusing on the transitions on the diagonal our technique can be easily adopted as an online processing method. In Fig 15 we showed the accumulated event frequency time series of the three experimental conditions, demonstrating the online-adopted analysis. This figure shows that the three conditions follows the same quantitative relations as the statistical results shown in Fig 14, which could be used to identify different visual-motor states.

## Conclusion

In this study, we applied the eigenvector-based dynamical network analysis method on the EEG functional connectivity, and successfully distinguished different conditions in visual-motor coordination by studying their meta-stable states dynamics. The dynamical method based on eigenvectors was first tested with simulated phase-locking networks, and the results showed that the inner product time series was able to characterize the member changes of clusters in the network. Second, our dynamical method was applied on the EEG data sets of a

visual-motor behavioural experiment. Both standard topological analysis and our dynamical method were able to distinguish different experimental conditions. In the alpha band, all the three topological properties (mean network density, mean cluster coefficient, and mean efficiency) were found significantly different between the Tra and VO condition, which indicates that the alpha band connectivity was related to the motion control part of the visual-motor coordination. In the gamma band, all the topological properties showed significant difference between the Tra and MO condition, which may imply that gamma band functional connectivity was related to visual input processing and the intention of tracking.

With the eigenvector-based dynamical analysis, we demonstrated the existence of a meta-stable states structure underlying the network dynamics, and could inspect the instantaneous transition of the prime eigenvector between different attractors. We found that the transition frequencies of the prime eigenvector were different across different experimental conditions. In the alpha band, the transition of prime eigenvectors were less often when participants were not involved with motor coordination, while the gamma band showed more frequent transition between different attractors, when participants performed the MO trials in which they observed the tracer, but were not engaged with any tracking process.

Our results showed that in the alpha and gamma band, the transition frequency of prime eigenvector between different meta-stable states could distinguish the characteristics of participant behavioural conditions as well as static topological properties. Thus, our results imply that there should be a significant correlation between the static and dynamical properties in the functional connectivity.

This study gives rise to new questions. Currently, the dynamical analysis method could only monitor dynamics of the largest cluster in the network using the prime eigenvector. In future works, we will study the other eigenvectors, and monitor evolution of minor clusters in the network. Also, we will further explore the attractor structure in prime eigenvector space, identify individual attractors, and reconstruct a full picture of the meta-stable state space.

## Supporting information

**S1 File. Parameter sweep of threshold.** In the Supplementary Information, we performed systematic analysis for a range of threshold to define functional connectivities in the alpha and gamma band.
(PDF)

## Acknowledgments

We would like to thank W. Harwin, P. Tolson and K. Djotni for development and maintenance of the haptic devices, and M. Baptista for fruitful discussion.

## Author Contributions

**Conceptualization:** Xinzhe Li, Bruno Mota, Slawomir Nasuto, Yoshikatsu Hayashi.

**Data curation:** Xinzhe Li.

**Formal analysis:** Xinzhe Li.

**Investigation:** Xinzhe Li.

**Methodology:** Xinzhe Li, Bruno Mota, Toshiyuki Kondo, Yoshikatsu Hayashi.

**Project administration:** Xinzhe Li, Yoshikatsu Hayashi.

**Software:** Xinzhe Li.

**Supervision:** Bruno Mota, Toshiyuki Kondo, Slawomir Nasuto, Yoshikatsu Hayashi.

**Validation:** Xinzhe Li.

**Visualization:** Xinzhe Li.

**Writing – original draft:** Xinzhe Li.

**Writing – review & editing:** Xinzhe Li, Bruno Mota, Toshiyuki Kondo, Slawomir Nasuto, Yoshikatsu Hayashi.

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
