## [Decision Letter · Decision Letter 0]

6 Nov 2019

PONE-D-19-27344

EEG Dynamical Network Analysis Method Reveals the Neural Signature of Visual-Motor Coordination

PLOS ONE

Dear Dr. Li,

Thank you for submitting your manuscript to PLOS ONE. After careful consideration, we feel that it has merit but does not fully meet PLOS ONE’s publication criteria as it currently stands. Therefore, we invite you to submit a revised version of the manuscript that addresses the points raised during the review process.

We would appreciate receiving your revised manuscript by Dec 21 2019 11:59PM. To enhance the reproducibility of your results, we recommend that if applicable you deposit your laboratory protocols in protocols.io, where a protocol can be assigned its own identifier (DOI) such that it can be cited independently in the future. For instructions see: http://journals.plos.org/plosone/s/submission-guidelines#loc-laboratory-protocols

We look forward to receiving your revised manuscript.

Kind regards,

Dimitris Kugiumtzis

Academic Editor

PLOS ONE

Journal Requirements:

We would like to thank Peter Tolson and Karim Djotni for development and maintenance of the haptic device. BM is acknowledges support by the Serrapilheira Institute (Serra-1709-16981) and CNPq (PQ 2017 312837/2017-8)

This study is an unfunded study.

5.  We note that Figure 1 includes an image of a [patient / participant / in the study]. 

Additional Editor Comments:

The reviewers raised serious and concrete issues with your submission, which you should address in your revision. In particular, in your revised manuscript you should:

1) Make clear whether you propose a new method and in that case stress what is the innovation added to this already in the literature. In particular, there are serious doubts of both reviewers, which I share, that your method of the eigenvector-based analysis is not novel.

2) Present clearly the advantages of the proposed method over other methods and show these in the simulations. In particular, point exactly to the results that show that it performs better (reviewer 1). Explain and address the issues regarding the design of the simulation study (reviewer 2).

2) If the contribution is focused on the application, give more convincing evidence for the relevance of your method(s) to the problem of the application, and in particular the estimated network evolution correlated with the nature of visual-motion coordination.

4) Improve the language.

Reviewers' comments:

Reviewer's Responses to Questions

**Comments to the Author**

1. Is the manuscript technically sound, and do the data support the conclusions?

Reviewer #1: Partly

Reviewer #2: Partly

2. Has the statistical analysis been performed appropriately and rigorously? 

Reviewer #1: Yes

Reviewer #2: No

3. Have the authors made all data underlying the findings in their manuscript fully available?

Reviewer #1: Yes

Reviewer #2: Yes

4. Is the manuscript presented in an intelligible fashion and written in standard English?

Reviewer #1: Yes

Reviewer #2: Yes

5. Review Comments to the Author

Reviewer #1: The main claim of the paper is the new method for analysis of the dynamic evolution of the functional neural networks, which is an important research question. The authors demonstrate the proposed method using simulated data and compare it to several other dynamic clustering methods. The authors also apply the proposed method to experimental EEG data set collected for visual-motor experiments. The authors claim that their method reveals the dynamic evolution of the functional neural networks during these experiments.

The introduction to the paper is written well, although it does not to mention adequately paper [40] (mentioned later elsewhere in the article), which seems to contain a method directly related to the method proposed here.

Looking at the results, the simulation shows that indeed the proposed method can pinpoint the time of dynamic changes in the connectivity of neural networks. However, the evidence for the proposed method showing an advantage over the dynamical clustering methods is not convincing. In fact, looking at the results presented in Fig.3-7, the competing methods seem to provide more information, i.e., not only a time point of change in the main cluster but also the number of clusters and their evolution over time. Both the proposed method and the ones compared demonstrate some minor issues. Disappointingly, it also seems that the proposed method is limited to finding the point in time when the dynamic change in the connectivity of neural network takes place, with the standard methods being applied for topological analysis of the networks as shown in Fig.9. Thus, I cannot agree that the proposed method, as it is explained and presented in the paper, is worthy of publication. Perhaps, if there were other advantages to the proposed method and they are highlighted clearly, it would be a different matter.

The article also presents the results of the standard topological analysis along with the newly proposed change detection method on the visual-motor EEG data. This analysis may well be worthy of publication, but the emphasis of the paper needs to be considered carefully. Is this a paper about analysis of the visual-motor EEG data using quite standard methods or is it a paper about a new method for the analysis of the dynamic evolution of neural network? If it is the former, then the emphasis of the paper should be on that. If it is the latter, then the advantages of the proposed method over the competing ones need to be explained better.

There are some other issues with the paper which are listed below. The abstract is too general and needs to be more specific about what kind of analysis is provided by the new method, i.e., what its output is. Reference [32] is very general describing many dynamical clustering methods. Explain why these particular methods were chosen for comparison. Substitute [32] for another reference if necessary. In Section "Eigenvector based dynamical analysis" explain how the values of different constants were chosen. In Fig.9, indicate where (a) and (b) are and increase font size in Fig.9-10. Add a reference for standard topological analysis in Section "Topological analysis". There are some minor typos in the article, which need to be corrected.

Reviewer #2: In this paper, Li and colleagues present a method to infer dynamical

transitions in network connectivity, and apply it to simulated and experimental

data. In simulation, the proposed method is able to detect changes in coupling

between the simulated oscillators. In a visuomotor experiment involving a

simple tracking task, the method detects some differences between tracking,

motor-only, and visual-only conditions. While my expertise is not on motor

neuroscience, and I am therefore unable to put these results into a specialised

neuroscientific context, I am confident in my assessment of the statistical and

computational aspects of the work.

The paper is interesting, and the method seems to work well in simulation.

However, I have three major concerns: (1) The experimental paradigm used is not

suitable to evaluate the proposed method; (2) the proposed method (or very

similar variants of it) have been presented before in the literature; and (3)

key elements of the technical description in the paper are ambiguous or

misleading, in a way that casts doubt on the interpretation of the results.

These concerns, in addition to a few minor comments and suggestions for

improvement, are described in more detail below.

## Major concerns

1) Fundamentally, the experimental paradigm used is an ill-suited application

of the proposed method. The method is presented as a technique to detect

dynamical transitions in network synchronisation, but *there are no transitions

in the experimental paradigm*. Subjects are instructed to track a target in

continuous circular motion -- what kind of transition do we expect to see here?

The choice of experimental paradigm is odd and, if the authors had any

particular hypothesis for investigating synchronisation transitions in this

setting, it should be clearly stated. The paper would make a much stronger case

if the key results related to the proposed method (that there are some

differences in transition frequencies between TRA, MO, and VO) were more

adequately interpreted and put into context.

While the method is interesting and potentially useful, I fail to see how it

reveals any neural signature of visuomotor coordination in the context of this

particular experimental paradigm. Describing precisely what the underlying

hypothesis about brain function was, and how the method sheds light on that

hypothesis, would make the paper much more accessible to the neuroscience

community.

2) Using eigenvectors to find structure in networks in a well-known field of

research. More than 10 years ago, it was already explored in the work of Newman

[1] and many other network scientists. More recently, and more directly

relevant to this paper, this technique was applied by Cabral and colleagues [2]

to the analysis of dynamical connectivity of fMRI data. Cabral's method is

virtually identical to the one presented here: compute the phase of the Hilbert

transform of each channel, compute some form of functional connectivity (phase

coherence in Cabral's work, pseudo-PLV here), and take the inner product

between leading eigenvectors of these time-varying connectivity matrices.

The authors need to make clear what the novel elements of their method are, and

cite the relevant literature. If there are any new elements in the proposed

method, they should be clearly stated, and the benefits of introducing them

should be assessed. If there aren't, the authors should spend less space

explaining the method and simply refer to the literature.

3) There are four major statistical or analysis errors that cast doubt on the

validity of the results. They are listed here in no particular order:

- The study claims to study synchronisation in 8 Hz oscillations using a window

of 40ms. *That is less than a third of a single period* of the oscillation,

so phase-locking values are likely to have a very strong bias. This is also

an issue (although less dramatic) for the other frequency bands. The authors

should take this bias into account for their analyses, as well as investigate

(and report) the effect of different window sizes.

- The results of the topological analysis in Fig. 8 do not seem properly

controlled. Naturally, a denser network will have both a higher clustering

coefficient and shorter path lengths -- not necessarily because of

fundamentally different properties of the network-generating process, but

*simply because the network is denser*. To make the findings interpretable

and unambiguous, the results for clustering coefficient and efficiency should

be compared against suitable null models (for example, random Erdos-Renyi

networks of the same density). Also, there is no mention whatsoever of

corrections for multiple comparisons, which need to be applied when

conducting an analysis on several measures in several frequency bands.

- In line 281, the authors say that the matrix was binarised "with a

visual-based threshold." I can't emphasise enough how important the

consequences of this threshold choice are. There is also an extensive

literature on the impact of such choices, see for example Ref. [4] among many

others. If thresholding is absolutely necessary, then authors should conduct

extensive parameter sweeps and controls to ensure that the results are

genuine and not an artefact of the "visual-based threshold."

- The quantity in Eq. (4) is not an actual phase-locking value. The PLV is

related to the average difference between the phases (as correctly described

in Eq. (1)), while Eq. (4) i) takes the difference between complex

exponentials instead of raw phases, and ii) takes the square of these

differences for each timestep. Both of these could dramatically change the

behaviour of the measure. Why not simply take the actual PLV, using the

standard formula?

PLV(m,n) = (1/T) | \\sum_{t=1}^T e^{i (\\phi_{m,t} - \\phi_{n,t}) } |

This would make the paper's results much easier to interpret, and it would be

much less confusing for readers. The dynamical connectivity matrix can be

defined using the same sliding window method that is currently used, and the

method should be just as applicable.

## Minor comments

- Overall, the language in the paper is understandable and gets the point

across, but the paper could substantially benefit from some more thorough

language editing. Text is excessively verbose, and it takes the reader a

long time to get through simple concepts. There are occasional typos and

misused articles.

- I recommend the term "prime eigenvector" introduced by the authors is

replaced by the standard nomenclature of "leading eigenvector."

- The details of the simulations could be more clearly specified. For example

the coefficients a_{m,n} or the initial angular velocities \\dot{\\phi}_{n,0}

are not specified. Also, given that this is a non-standard model, it would

be very informative for the reader to see some time series of the phases

themselves to build a better mental image of the system. Alternatively, the

authors could use a more well-known oscillator model, such as coupled

Kuramoto oscillators. For example, one particularly attractive domain of

application would be metastable chimera states [4].

- In the discussion of analysis results in line 564, what does it mean for "the

frequencies of 0 event and 1 event" to be "highly consistent with the

topological properties"? The topological properties make no statement about

network dynamics, and no null model or hypothesis is presented to compare

those two.

- Regarding the financial disclosure: please write, as indicated in the

submission guidelines, "The authors received no specific funding for this

work."

- The quality of the figures could be substantially improved. They are very

clearly the default Matlab settings, which are not of publication quality.

There are also a few particular issues in some figures, for example: in Figs.

3 and 4, it is very confusing that the X-axis for two subplots is time, but

in the third plot it is entry index; Fig. 7 has a continuous colour bar but

discrete values; Fig. 9 has tiny fonts and no subfigure labelling.

- Given that PLoS ONE does not apply any copyediting to the paper, the authors

should spend significant effort on it (for example, positioning of figures

5,6,8, font sizes in figures 9,10).

[1] Newman, M. E. Finding community structure in networks using the eigenvectors of

matrices. Physical Rev. E 74, 036104 (2006) doi:10.1103/PhysRevE.74.036104

[2] Cabral, J., Vidaurre, D., Marques, P. et al. Cognitive performance in healthy

older adults relates to spontaneous switching between states of functional

connectivity during rest. Sci Rep 7, 5135 (2017) doi:10.1038/s41598-017-05425-7

[3] Drakesmith, M., K. Caeyenberghs, K., Dutt, A., Lewis, G., David, A.S., and

Jones, D.K. Overcoming the effects of false positives and threshold bias in

graph theoretical analyses of neuroimaging data. NeuroImage 118, 313-333 (2015)

doi:10.1016/j.neuroimage.2015.05.011

[4] Shanahan, M. Metastable chimera states in community-structured oscillator

networks. Chaos 20, 013108 (2010) doi:10.1063/1.3305451

6. PLOS authors have the option to publish the peer review history of their article (what does this mean?). If published, this will include your full peer review and any attached files.

Reviewer #1: No

Reviewer #2: No

---

## [Author Response · Author response to Decision Letter 0]

6 Feb 2020

Please see Response to Reviewers letter at the end of this PDF

---

## [Decision Letter · Decision Letter 1]

1 Apr 2020

EEG Dynamical Network Analysis Method Reveals the Neural Signature of Visual-Motor Coordination

PONE-D-19-27344R1

Dear Dr. Li,

We are pleased to inform you that your manuscript has been judged scientifically suitable for publication and will be formally accepted for publication once it complies with all outstanding technical requirements.

With kind regards,

Dimitris Kugiumtzis

Academic Editor

PLOS ONE

Additional Editor Comments (optional):

Reviewers' comments:

Reviewer's Responses to Questions

**Comments to the Author**

1. If the authors have adequately addressed your comments raised in a previous round of review and you feel that this manuscript is now acceptable for publication, you may indicate that here to bypass the “Comments to the Author” section, enter your conflict of interest statement in the “Confidential to Editor” section, and submit your "Accept" recommendation.

Reviewer #1: All comments have been addressed

Reviewer #2: (No Response)

2. Is the manuscript technically sound, and do the data support the conclusions?

Reviewer #1: Yes

Reviewer #2: Partly

3. Has the statistical analysis been performed appropriately and rigorously? 

Reviewer #1: Yes

Reviewer #2: Yes

4. Have the authors made all data underlying the findings in their manuscript fully available?

Reviewer #1: Yes

Reviewer #2: Yes

5. Is the manuscript presented in an intelligible fashion and written in standard English?

Reviewer #1: Yes

Reviewer #2: Yes

6. Review Comments to the Author

Reviewer #1: The authors addressed most of the issues raised by myself. If the other reviewers are also satisfied with how their comments were addressed, the paper can be accepted.

Reviewer #2: Thanks to the authors for revising the manuscript and replying to our earlier

comments. I appreciate that most of the comments have been resolved, in

particular those regarding font sizes in the figures, simulation and analysis

details, and relation to prior work. While the eigenvector method itself is not

fully novel (and, accordingly, it is not presented as such in the paper), the

analyses of simulated and real data seem valid and worthy of publication.

I have one particular remaining comment about the threshold: in the revised

manuscript, the authors describe how they chose their visual-based threshold.

This is useful, although it does not in any way address my earlier comment of

how sensitive the results can be to this choice (and no parameter sweep was

carried out). This could potentially have a big impact in the results (for

example, in the example in Fig. 1 a slightly higher threshold could mean P1 and

PZ remain "phase-locked" for the entire duration of the example).

I believe that, although the paper is full of idiosyncrasies and somewhat

arbitrary analysis choices (such as using the Euclidean distance between

angular speed vectors instead of normal PLV), these choices are described in

detail, and therefore the paper meets the publication criteria at PLoS ONE.

Although it is not a strict requirement, I have a few additional minor comments

that I would strongly recommend the authors to address for the final form of

the manuscript:

- It would be really helpful if the subplots in Fig 10 would include a vertical

line indicating the average, which is typically the quantity of interest.

Related to this, I suspect there may be an important typo in line 544, which

should say "lower efficiency *than* in the random network" (since the

efficiency in the EEG network is visibly lower than in the random network).

- The threshold discussion above might have a negative effect on readers, and

hinder the authors' credibility. I would still recommend the authors perform

a parameter sweep and check the results hold for a broad range of thresholds,

perhaps including these results as supplementary material.

7. PLOS authors have the option to publish the peer review history of their article (what does this mean?). If published, this will include your full peer review and any attached files.

Reviewer #1: No

Reviewer #2: No

---

## [Editor Report · Acceptance letter]

15 May 2020

PONE-D-19-27344R1 

EEG Dynamical Network Analysis Method Reveals the Neural Signature of Visual-Motor Coordination 

Dear Dr. Li:

I am pleased to inform you that your manuscript has been deemed suitable for publication in PLOS ONE. Congratulations! Your manuscript is now with our production department. 

With kind regards,

on behalf of

Prof Dimitris Kugiumtzis 

Academic Editor

PLOS ONE